# Control of metal oxides' electronic conductivity through visual intercalation chemical reactions

Yuanyuan Zhang[1], Xiaohua Zhang [2], Quanquan Pang [3] & Jianhua Yan [1,2,4] ✉

Cation intercalation is an effective method to optimize the electronic structures of metal oxides, but tuning intercalation structure and conductivity by manipulating ion movement is difficult. Here, we report a visual topochemical synthesis strategy to control intercalation pathways and structures and realize the rapid synthesis of flexible conductive metal oxide films in one minute at room temperature. Using flexible $TiO_2$ nanofiber films as the prototype, we design three charge-driven models to intercalate preset $Li^+$-ions into the $TiO_2$ lattice slowly (μm/s), rapidly (mm/s), or ultrafast (cm/s). The $Li^+$-intercalation causes real-time color changes of the $TiO_2$ films from white to blue and then black, corresponding to the structures of $Li_xTiO_2$ and $Li_xTiO_{2-\delta}$, and the enhanced conductivity from 0 to 1 and 40 S/m. This work realizes large-scale and rapid synthesis of flexible $TiO_2$ nanofiber films with tunable conductivity and is expected to extend the synthesis to other conductive metal oxide films.

Intercalation chemical reaction is a basic and important chemical reaction, which has always dominated the fields of electrochemical energy storage. Correspondingly, the intercalation chemistry-based transition metal oxides (ITMO) have important applications in electrical energy storage[1–3]. However, most ITMO exhibit slow cationic diffusion and poor electronic conductivity that limit their performance[4,5]. Various methods have been developed to tune the electronic structures of ITMO[6–9]. Among them, cation doping is an effective strategy to synthesize new structures with metastable lattice environment and different valence states, which takes advantage of the unbalanced kinetics of different cations[10]. ITMO such as $TiO_2$, niobium pentoxide and vanadium pentoxide have opening tunnel networks and abundant states of valence for the metal, which make them good platforms for guest species intercalation to modulate ITMO properties. The degree of freedom of the electronic structures can be tuned by doping different amounts of cations[11].

Traditional strategies for cation doping to tune the ITMO structures include solid-state reaction, hydrothermal/solvothermal treatment, and ionic heat treatment, but it is difficult to design and obtain target electronic structures with these methods[12–15]. Recently, numerous studies have been reported on the intercalation of alkali metal (lithium, sodium, potassium) cations into the host ITMO materials to engineer their electronic properties[16–18]. For example, $Li^+$-ions can be intercalated into $TiO_2$ to form metastable lithium titanate, an attractive anode that exhibits extraordinary rate capability and shows great prospects for designing fast-charging $Li^+$-ion batteries[16]. These studies indicate that using electrochemical intercalation reactions is a feasible strategy to adjust the electronic structures of ITMO. However, there are still some problems. For one thing, the intercalation chemical reaction is like a black box, and due to the lack of ability to control the intercalation thermodynamics and kinetics, which are considered as the main factors controlling the intercalation reactions, it is difficult to intuitively observe and manipulate the ion movement and intercalated structures[19–21]. Moreover, the regulation mechanism of metal oxides' electronic conductivity by intercalation structures is far to be understood.

Here, we report a topochemical synthesis strategy (namely intercalation paths) to visualize the real-time synchronous cation

[1]College of Textiles, Donghua University, 201620 Shanghai, China. [2]Innovation Center for Textile Science and Technology, Donghua University, 200051 Shanghai, China. [3]Beijing Key Laboratory of Theory and Technology for Advanced Batteries Materials, School of Materials Science and Engineering, Peking University, 100871 Beijing, China. [4]State Key Laboratory for Modification of Chemical Fibers and Polymer Materials, Donghua University, 201620 Shanghai, China. ✉e-mail: yanjianhua@dhu.edu.cn

transport and electron conduction pathways and realize the control of intercalation structures and metal oxides' conductivity. This proof-of-concept is studied by using a flexible topological TiO₂ nanofiber (NF) film as the prototype and designing three types of Li⁺-gated, electron-gated, and Li⁺/electron co-gated charge-driven models to advance the preset Li⁺-ions to intercalate into TiO₂ lattices slowly (μm/s), rapidly (mm/s) or explosively (cm/s). Both the experiments and calculations show that the initial concentrations of the resident charges on TiO₂ NFs determine the electron conduction paths, which then establish the Li⁺-intercalation paths. The Li⁺-intercalation in all models led to the reduction of $Ti^{4+}$ to $Ti^{3+}$, lattice expansion, and the creation of oxygen vacancy in TiO₂ crystals, resulting in real-time color changes of the film from white to blue and then black, and the synchronous intercalation-based build-up of electron conduction pathways. Both the color and conductivity are closely related to the intercalated structures, which contained a low stable but high conductive black $TiO_{2-\delta}$ structure (>40 S/m) and a high stable but low conductive blue $Li_xTiO_2$ structure (1–40 S/m). The conductivity of TiO₂ can be facilely tuned by controlling the intercalation process. Unlike previous studies on ITMO powders, the real-time intercalation pathways were observed and

controlled. Importantly, this strategy is expected to extend to other cation intercalations and oxides for the synthesis of conductive metal oxide films on a large scale at room temperature.

## Results

### Topochemical synthetic conductive black TiO₂ NF films at room-temperature

Figure 1 shows the schematic diagram of topochemical synthetic conductive black TiO₂ NF films at room temperature based on three charge-driven models. The first model is shown in Fig. 1a. In this model, a piece of white TiO₂ NF film is put on a Li-metal sheet, and the interface is infiltrated by a drop of dimethylacetamide (DMAc) solvent. The highly active Li-metal can quickly react with the TiO₂ at the interface and create active Li-nanoparticles (NPs), solvated electrons ($e_s^-$) and Li⁺@DMAc. The DMAc solvent essentially acts as a phase transfer reagent that continuously and rapidly transfer these active species into the film due to the Siphoning effect of the nano-porous film structure (Supplementary Fig. 1). The active NPs then react with the other TiO₂ NFs layer-by-layer, while the solvated Li⁺ intercalate into TiO₂ to jointly form black $Li_xTiO_{2-\delta}$ ($0 \le x \le 1$; $0 \le \delta \le 2$), where $x$ and $\delta$ represent the

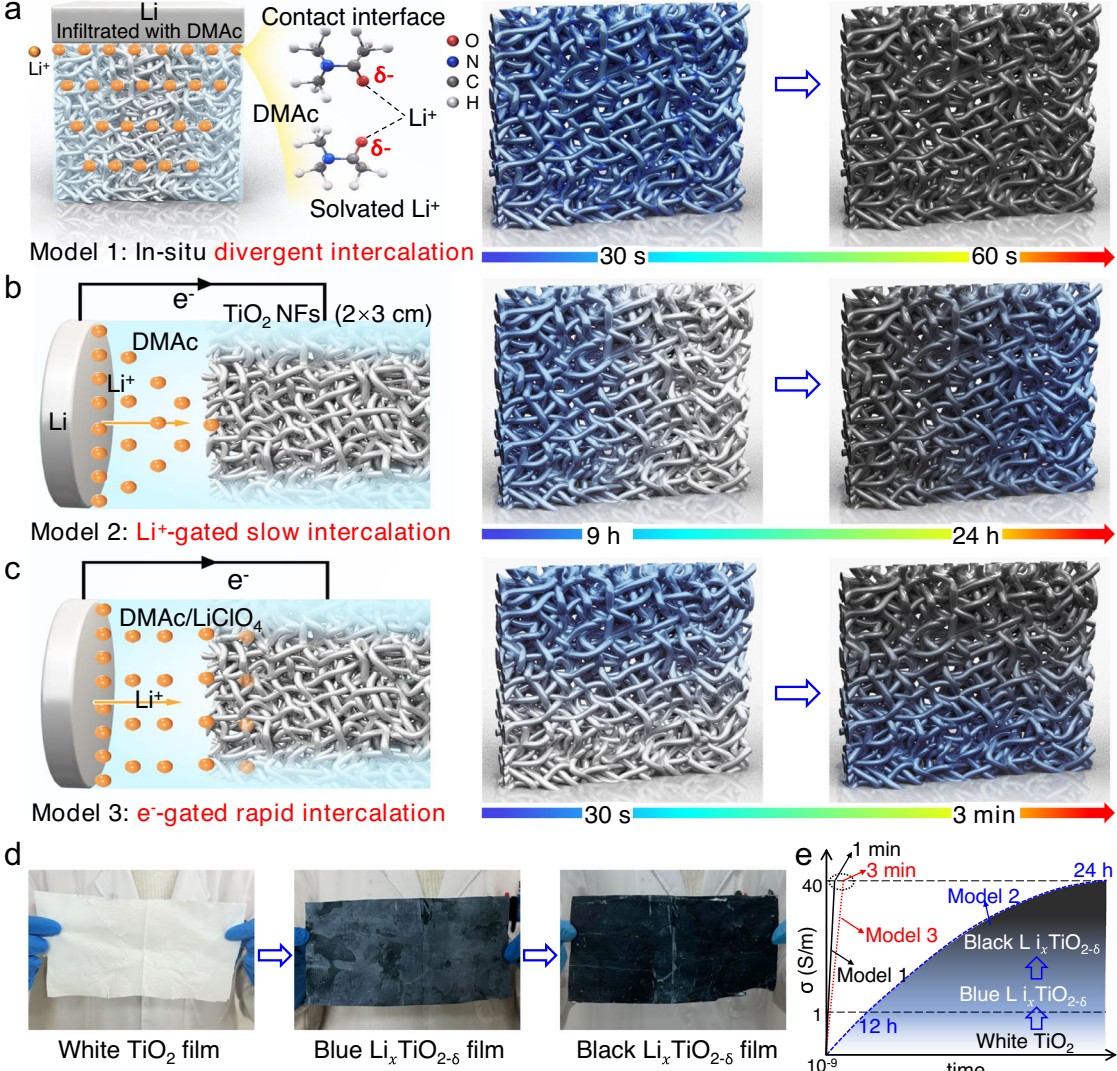

**Fig. 1 | Topochemical synthetic conductive black TiO₂ NF films at room-temperature. a** The in situ divergent intercalation model and the synchronous color changes of the TiO₂ NF film. **b** The Li⁺-gated slow intercalation model and the color changes of the film along the Li⁺-diffusion from left to right. **c** The electron-

gated rapid intercalation model and the color changes of the film along the electron conduction from top to down. **d** A large piece of white TiO₂ NF film and the blue and black $Li_xTiO_{2-\delta}$ NF films treated by model 3. **e** The evolution of color and conductivity of the TiO₂ NF film with intercalation depth (time).

intercalation intensity and the concentration of oxygen vacancy. As a result, the white $TiO_2$ NF film turns blue and then black within 1 min along the direction perpendicular to the interface (Supplementary movie 1), and its conductivity increases sharply from 0 to >40 S/m. This model is named as an in situ divergent intercalation reaction since both the electrons and $Li^+$-ions transfer divergently from the interface and establish divergent intercalation paths. We found that the changes in color and conductivity of the white $TiO_2$ NFs were caused either by the $Li^+$-intercalation induced reduction of $Ti^{4+}$ into $Ti^{3+}$ or the deprivation of lattice oxygen by Li-metal[22,23]. However, the specific roles of these two mechanisms are not clear.

To make it clear, another two models are designed. The 2nd model is shown in Fig. 1b. In this model, a white $TiO_2$ NF film (2 cm × 3 cm) and a circular Li-metal film ($d = 2.1$ cm) are vertically outward immersed into the DMAc solvent, and then connected by a wire on the top (Supplementary Fig. 2a, b). The distance between these two films is set as 10 cm. The solvated $Li^+$@DMAc will transfer from the left Li-metal side to the right NF film side due to the concentration difference, and a continuous $Li^+$-ion diffusion channel and a battery circuit are formed after 9 h. At this time, electrons will flow to the same place where the $Li^+$-ions have just reached and initiate the intercalation reaction. This reaction advances gradually along the $Li^+$-diffusion from left to right. As a result, the white $TiO_2$ NF film began to turn blue from the left side after 9 h, and it took another 15 h to turn black. Interestingly, the left side of the $TiO_2$ NF film was darker, indicating that more $Li^+$-ions had been intercalated into the left side and the intercalation intensity influenced the color. This model is named as a $Li^+$-gated slow intercalation reaction due to the slow $Li^+$-diffusion. The NF film still showed a blue color and a small conductivity of ~1 S/cm after 3 h intercalation (Supplementary Fig. 2c), and a black color and a high conductivity of ~40 S/m after 24 h intercalation.

If adding some salt ($LiClO_4$) into the DMAc solvent and maintaining the other conditions unchanged (here is the third model, Fig. 1c and Supplementary Fig. 3a), the white $TiO_2$ NF film quickly turned black from the top to the bottom within 3 min (Supplementary movie 2). The detail for the color change process of the white $TiO_2$ NF film is shown in Supplementary Fig. 3b. In this model, many $Li^+$-ions resided on the NF surface beforehand since there were sufficient $Li^+$-ions in DMAc, and the intercalation immediately started when connecting these two films with a wire. As a result, the intercalation reactions occurred along the electron conduction, thus forming a reaction channel gated by electrons. Since the electron conduction was far greater than $Li^+$-ion diffusion, the film quickly turned blue and black along the electron conduction paths from top to down, and this model is named as an electron-gated rapid intercalation reaction. Correspondingly, the intercalated $Li_xTiO_{2-\delta}$ film showed a black color (Supplementary Fig. 3c) with a high electronic conductivity of ~40 S/m after 3 min intercalation.

Figure 1d shows a large piece of flexible $TiO_2$ NF film (10 cm × 25 cm) and the resulting conductive blue (1–40 S/m) and black (>40 S/m) $Li_xTiO_{2-\delta}$ NF films based on the 3rd model. From these three models, we found that the degree of color changes in the $TiO_2$ NF film was proportional to the intercalation intensity and electronic conductivity. All the black $Li_xTiO_{2-\delta}$ NF films had a high conductivity of >40 S/m and the blue $Li_xTiO_{2-\delta}$ NF films had a low conductivity of 1–40 S/m. The evolution of color and conductivity of the NF film with intercalation depth (time) is shown in Fig. 1e.

## $Li^+$-intercalation pathways and the synchronous build-up of intercalation-based electron conduction paths in the $TiO_2$ NF films

As for the driving forces of $Li^+$-intercalation, the general understanding lies in that the intercalation is forced by either the solid-state physic mechanical force or the liquid-state chemical potential energy. As a host material, $TiO_2$ has open channels in the lattice and small $Li^+$-ions can

overcome lower barrier energy to intercalate into the lattice of $TiO_2$ at room temperature[24]. In model 1, the strong reductant Li-metal can directly reduce the $TiO_2$ in contact with it due to the difference in energy level barrier (Fig. 2a) and the created solvated active $Li/Li^+$ due to the interaction with DMAc can synchronously intercalate into the $TiO_2$ lattice[25]. Some solvents like DMAc that contains carbonyl-O functional group had the ability to pull or lock electrons to form active solvated $Li/Li^+$ and solvated $e_s^-$ and could form strong Coulombic interaction with $Li^+$-ions (Fig. 2b)[26]. These active species are highly reactive nucleophiles that can accelerate the reduction of $TiO_2$ NF film by Li-metal.

Relying on the advantages of the flexible topological $TiO_2$ NF film structure, the real-time conduction pathways of electrons and $Li^+$-ions can be visually observed by recording the gradual color changes along the NF film. The visual displays of experimental models 1–3 are shown in Fig. 2c–e. For each model, the initial and final states ($t = 1$ min for model 1, 24 h for model 2, and 3 min for model 3) are recorded. The intercalation reaction in model 1 occurs in a divergent manner with an explosive reaction rate (cm/s) since lots of $Li/Li^+$ and electrons synchronously reside on the $TiO_2$ NFs (Fig. 2c). In model 2, the limiting factor of the intercalation is the slow $Li^+$-diffusion. Since the electron conduction is far greater than the free diffusion of $Li^+$-ions, it can be regarded that many electrons have resided on the $TiO_2$ NFs to wait for $Li^+$-ions to arrive and then start the intercalation (Fig. 2d). Therefore, the intercalation shows a slow rate scale of ~μm/s. By contrast, in model 3, there are sufficient resident $Li^+$-ions on the $TiO_2$ NFs to wait for electrons to arrive and then start the intercalation (Fig. 2e), thus demonstrating a rapid intercalation rate scale of ~mm/s.

From the experimental phenomena, we can conclude that the synchronous conduction of electrons and ions is a necessary condition for intercalation, and the initial concentration of electrons and $Li^+$-ions that reside on the $TiO_2$ NFs first determine the electron transfer paths, which then establish the $Li^+$-intercalation paths. In addition, the intercalation rates are closely related to the conduction pathways of electrons and $Li^+$-ions along the $TiO_2$ NF films. The intercalation-based build-up of electron conduction pathways is shown in Fig. 2f–h, corresponding to the three different models. In model 1, the large resident electrons and $Li^+$-ions established divergent intercalation paths with an explosive intercalation rate. In model 2, due to the random $Li^+$-diffusion paths (reminiscent of the random walk-in graph theory) controlled by concentration difference, the electron conduction is roundabout, and the paths are not optimal, leading to a slow intercalation rate. In model 3, many $Li^+$-ions resided on the NF surface beforehand, which would form multiple parallel high-speed channels for electron conduction. Therefore, through different topological charge-driven models, the intercalation intensity and conductivity of $TiO_2$ can be adjusted.

## Characterization of the blue and black intercalation structures

Different intercalation pathways not only affect the intercalation structures but also affect the morphology of the $TiO_2$ NFs. As shown in Fig. 3a, the as-fabricated flexible white $TiO_2$ film is composed of countless intertwined smooth NFs, and its electronic conductivity is close to 0. The large-scale fabrication of flexible $TiO_2$ NF films by electrospinning and their fire-resistant property are described in Supplementary Fig. 4 and Supplementary movie 3. The polycrystalline electrospun $TiO_2$ NF is usually brittle due to its large grain size and pore defects[27]. To solve this problem, we ball milled the spinning sol to promote the formation of small grains and adopted a gradient calcination method to reduce the uncontrollable pore defects generated by polymer decomposition, thereby obtaining the flexibility of $TiO_2$ NFs. The intercalated conductive $Li_xTiO_{2-\delta}$ NF films in model 2 (Fig. 3b) and 3 (Fig. 3c) were characterized by scanning electron microscopy (SEM). Unlike the traditional high-temperature reduction strategy that easily led to brittle fracture of $TiO_2$ NFs, this room-temperature intercalation reaction maintained the flexibility of the NFs well. The NF surface

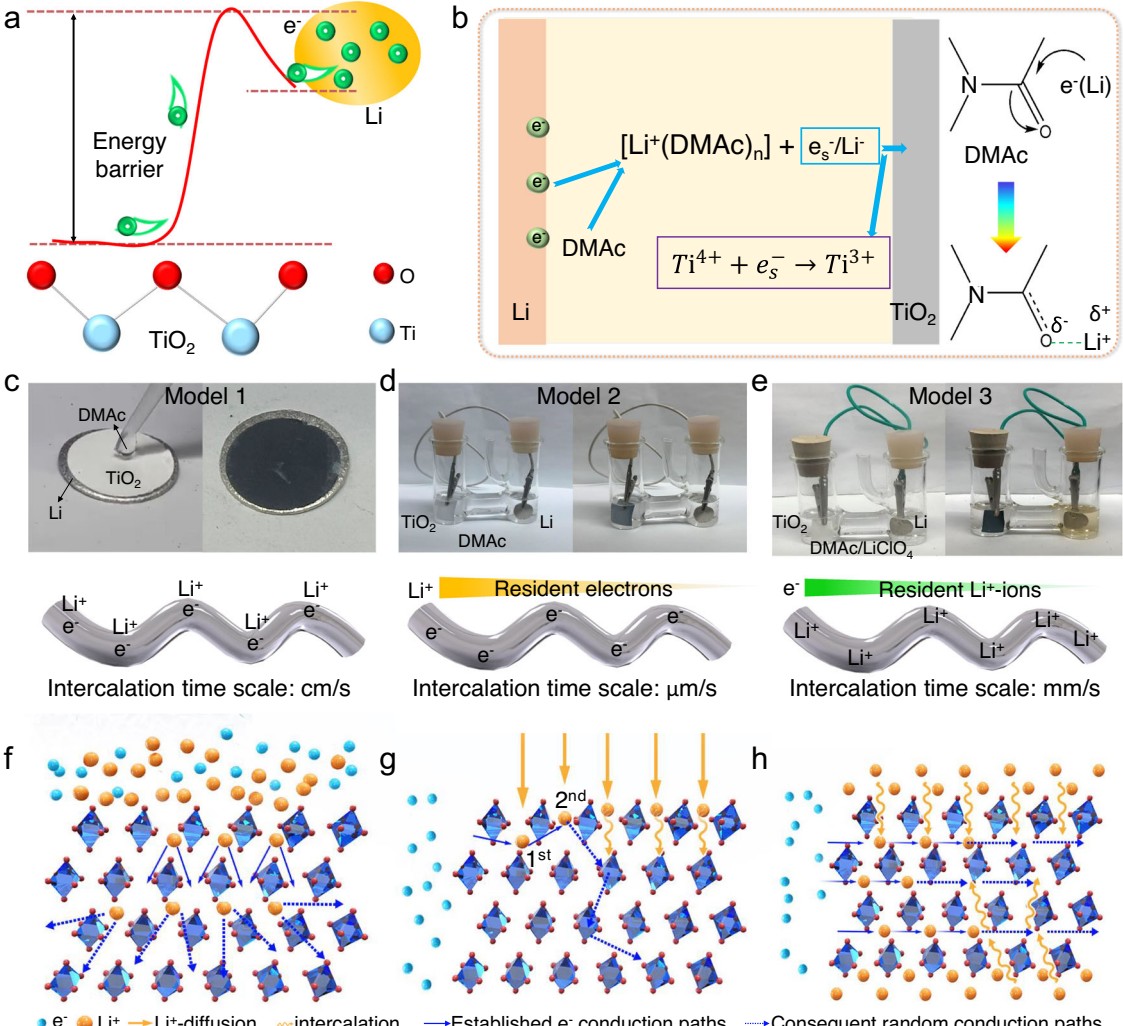

**Fig. 2 | Mechanism illustration of the intercalation pathways and the synchronous build-up of electron conduction paths. a** Schematic diagram of the movement of electrons between Li-metal and $TiO_2$. **b** Li$^+$-interaction kinetics in DMAc by forming solvated Li/Li$^+$. **c**–**e** The experimental models 1–3 and the schematic diagrams of resident charges on $TiO_2$ NFs. **f** Synchronous resident electrons and Li$^+$-ions resulted in an explosive intercalation path in model 1. **g** Random Li$^+$-ion diffusion led to a roundabout and slow Li$^+$-intercalation path in model 2. **h** A high concentration of Li$^+$-ion induced a fast and directional Li$^+$-intercalation path in model 3.

became rough and contained precipitated NPs caused by the formation of $Li_2O$. There were more precipitated NPs on the $TiO_2$ NFs in model 3 and this film showed a darker color change (named as D-Li$_x$TiO$_{2-\delta}$), indicating a higher intensive intercalation reaction.

To explore the correlation of the intercalated structures with color and conductivity, high-resolution transmission electron microscopy (HRTEM) images were recorded. The white $TiO_2$ NFs exhibited clear lattice fringes with an in-plane characteristic d-spacing of 0.352 nm (Fig. 3d), corresponding to the (101) lattice plane of anatase $TiO_2$[28]. The low conductive blue Li$_x$TiO$_{2-\delta}$ NFs showed a slight increase of lattice distance in comparison with the white $TiO_2$ (Fig. 3e), and the high conductive D-Li$_x$TiO$_{2-\delta}$ NFs exhibited a larger lattice distance of 0.361 nm (Fig. 3f). The changes of d-spacing can be observed more intuitively from the inverse fast Fourier transform (FFT) images (Fig. 3g–i). Both samples exhibited lattice distortions after the intercalation, but the D-Li$_x$TiO$_{2-\delta}$ showed a higher degree of distortion. Therefore, the Li$^+$-intercalation changed the electronic structure of $TiO_2$, and the lattice expansion increased with increasing the intercalation intensity.

However, according to X-ray diffraction (XRD) patterns, the diffraction peaks for both Li$_x$TiO$_{2-\delta}$ and D-Li$_x$TiO$_{2-\delta}$ corresponded to anatase $TiO_2$, indicating that no obvious phase transition was induced after the intercalation (Fig. 3j)[29]. Nevertheless, there were enlarged (101) peaks that shifted toward lower angles as compared with the white $TiO_2$, confirming that there was lattice expansion in $TiO_2$ crystals and the formation of oxygen vacancies[30]. X-ray photoelectron spectra (XPS) were conducted to further evaluate the intercalated structures. As shown in Fig. 3k, the spectra of O 1s showed three characteristic peaks that were attributed to lattice oxygen (O1), defects (O2), and chemisorbed oxygen species (O3), respectively[31]. Compared with the white $TiO_2$, the oxygen peak shifted to higher binding energy, and the oxygen vacancies determined by the area ratio of O2 peak increased after the intercalation, indicating a higher oxidation state of oxygen in Ti-O[32]. In addition, there were chemisorbed oxygen species for both Li$_x$TiO$_{2-\delta}$ and D-Li$_x$TiO$_{2-\delta}$ samples due to the existence of DMAc[33]. On the other hand, the Ti 2p$_{3/2}$ spectra (Fig. 3l) confirmed that the intercalation caused the reduction of Ti$^{4+}$ to Ti$^{3+}$, and the D-Li$_x$TiO$_{2-\delta}$ contained a higher Ti$^{3+}$ concentration than the Li$_x$TiO$_{2-\delta}$ sample.

These results confirmed that the Li$^+$-intercalation did not change the crystalline phase of $TiO_2$, but introduced lattice oxygen vacancies and Ti$^{3+}$ sites that distorted the lattice. In addition, the concentration of Ti$^{3+}$ increased with increasing the intercalation intensity, and the Ti$^{3+}$ species come from two different ways that caused the color changes. First, the Li$^+$-intercalation in $TiO_2$ can form Ti$^{3+}$ and result in the white

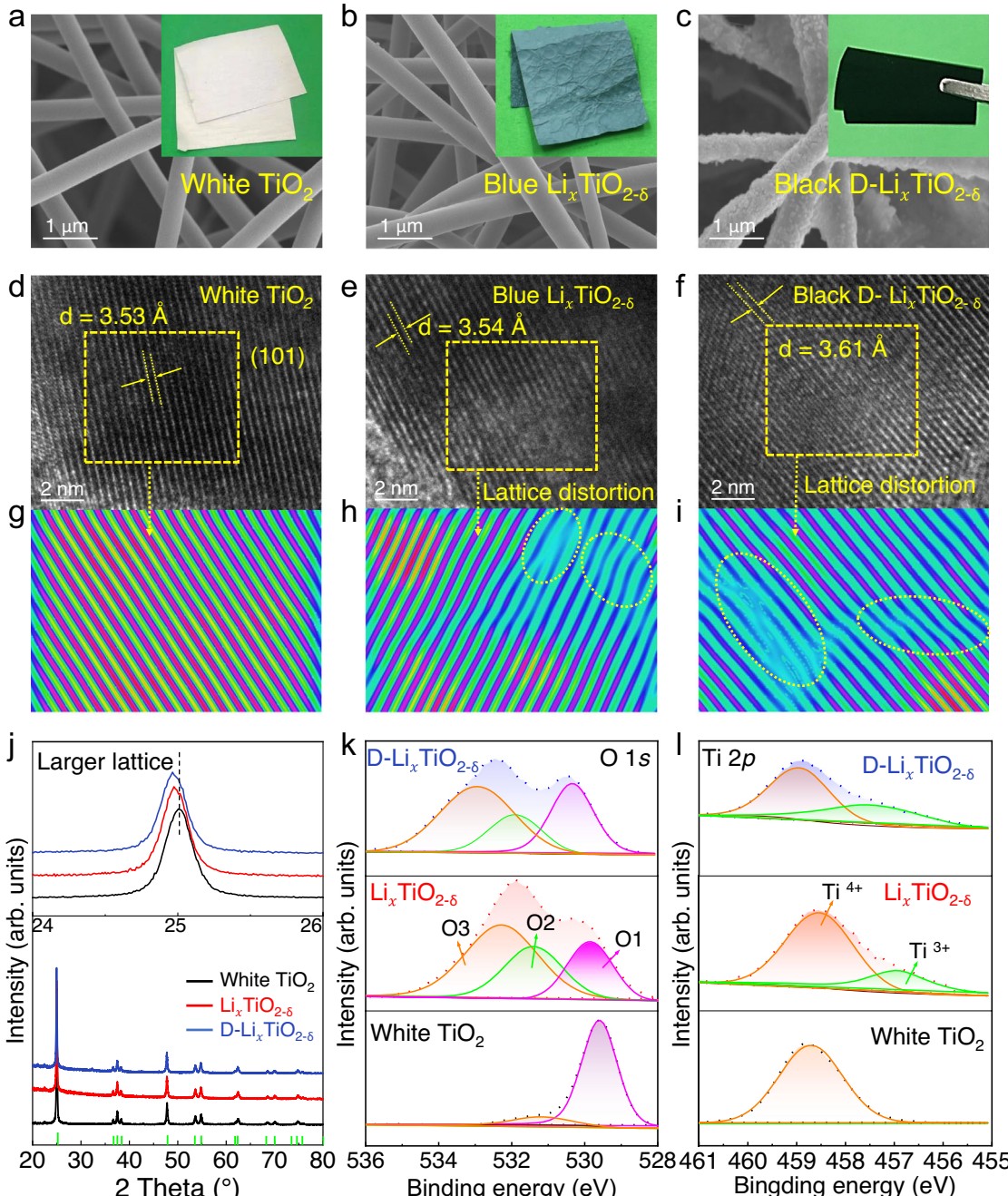

**Fig. 3 | Characterizations of the intercalated structures. a–c** SEM images and digital photos of the white $TiO_2$, $Li_xTiO_{2-\delta}$ (model 2) and D- $Li_xTiO_{2-\delta}$ (model 3) NF films. **d–f** TEM images, **g–i** Inverse FFT images, and **j** XRD patterns of the white $TiO_2$, and black $Li_xTiO_{2-\delta}$ and D-$Li_xTiO_{2-\delta}$ NF films. High-resolution XPS spectra of **k** O $1s$ and **l** Ti $2p_{3/2}$ of the three film samples.

to blue transformation of the $TiO_2$ NF film. Second, with the $Li^+$-intercalation reactions going on and once the intercalation structure is saturated, $Li^+$-ions will obtain electrons and deposit on the $TiO_2$ surface. These metallic Li can deprive the lattice oxygen to form vacancies, and $Ti^{3+}$ defects will thus be generated due to charge compensation, resulting in a transformation of the blue $TiO_2$ to black. Therefore, both the $Ti^{3+}$ and lattice oxygen vacancy can induce color change, and the oxygen vacancy-induced $Ti^{3+}$ species can significantly enhance the film conductivity[34].

## Conductive mechanisms of the blue and black $Li_xTiO_{2-\delta}$ NF films
The direct manifestation of $Li^+$-intercalation is that the intercalated structures showed significantly enhanced electronic conductivity.

According to the classical theory of electron conduction in crystals, electrons can conduct along the Ti-O bond or lattice oxygen vacancy. For the former, electrons on the highest band first jump to the adjacent empty band under an electric field and tunnel, the Ti-O bond once a threshold concentration is reached[35]. Accompanying changes include the conversion of $Ti^{4+}$ into $Ti^{3+}$, and the move forward of the remaining electrons to the next $Ti^{4+}$ center to reside. At the same time, $Li^+$-ions begin to intercalate and form conductive $Li_xTiO_{2-\delta}$ phases, which serve to pave a highway channel for the subsequent intercalation. For the latter, the oxygen vacancy with stoichiometric deviation from hypoxia of $TiO_{2-\delta}$ can form donor energy levels to promote carrier migration. Due to the introduction of oxygen vacancies defects, the Fermi energy of $TiO_{2-\delta}$ can move into the conduction band, thus exhibiting higher

conductivity[36]. Therefore, the black and blue $Li_xTiO_{2-\delta}$ NFs with vacancy defects show high conductivity.

The defective structures of $D\text{-}Li_xTiO_{2-\delta}$ and the tunneling of Ti-O bonds were then investigated by theoretical calculations. The simulation and calculation of the $Li^+$-ion adsorption on the $TiO_2$ lattice surface and the accompanying charge transfer pathways are shown in Fig. 4a. The adsorption model was set up based on the {101} facet of $TiO_2$ since the as-prepared anatase $TiO_2$ had a dominant facet of {101}[37]. The calculation results showed a clear charge transfer from the Li atom to the Ti atom, confirming that electrons can tunnel the Ti-O bond and then move forward to the next Ti-O bond to form a continuous conduction path along the Ti-O-Ti-O channel. The projected density of states (PDOS) of $TiO_2$ before and after adsorption of lithium is shown in Fig. 4b. The calculated PDOS plots exhibited that a new electron-filled band gap state consisting of Ti 3d sat on the left side of the Fermi Level, which explained the existence of the $Ti^{3+}$ species that owe to the partial reduction of Ti sites on the $TiO_2$ NF surface[38].

To further illustrate the structure of intercalated $TiO_2$ and the kinetics associated with $Li^+$-ions migration, kinetics simulations on the lithiation of $TiO_2$ were carried out. Figure 4c, d are the crystal structures after the $Li^+$-ions insertion and diffusion. After the incorporation of $Li^+$-ions, the $D\text{-}Li_xTiO_{2-\delta}$ containing oxygen vacancies demonstrated a smaller band gap (2.23 eV) as compared with the $Li_xTiO_2$ (1.136 eV), but both were smaller than pristine $TiO_2$ (Fig. 4e). The narrow band gap of $D\text{-}Li_xTiO_{2-\delta}$ suggests the reduction of Ti from +4 to +3 state and the enhanced electron transfer ability in $TiO_2$ induced by lithiation. Figure 4f exhibited the energy profiles along the transport pathway of $TiO_2$ after $Li^+$-ions insertion. The lithium migration barrier was calculated to be 0.62 eV for $TiO_2$ with a small amount of lithium inserted, which was higher than that of $D\text{-}Li_xTiO_{2-\delta}$ (0.48 eV). The low $Li^+$-ions diffusion barrier and excellent conductivity of $D\text{-}Li_xTiO_{2-\delta}$ indicate that it is a potential candidate anode for lithium-ion batteries.

On the other hand, the $Li^+$-intercalation intensity and oxygen vacancy concentration of the topochemical synthetic intercalated $D\text{-}Li_xTiO_{2-\delta}$ materials were characterized. As shown in Fig. 4g, the room-temperature electron paramagnetic resonance (EPR) spectra confirmed the high contents of $Ti^{3+}$ species and oxygen vacancies in the black intercalated structures. The g value of 2.003 in the black $D\text{-}Li_xTiO_{2-\delta}$ NFs was caused by oxygen vacancies, which was related to the formation of $Ti^{3+}\text{-}O\cdot$radical[39]. Besides, the Raman spectra of $D\text{-}Li_xTiO_{2-\delta}$ showed a similar configuration with the white $TiO_2$, but a clear blue-shift of peak and a broadened $E_g$ peak were observed, further confirming the robust intercalated $D\text{-}Li_xTiO_{2-\delta}$ structure and the structural evolution by nonstoichiometric measurements (Fig. 4h and Supplementary Fig. 5)[40].

### Structures and structure stability of the conductive blue and black $Li_xTiO_{2-\delta}$

From the above results, the conductivity of the intercalated NF film can be manipulated by controlling the intercalation pathways and intensity, but the intercalation structure is still unclear. To determine the intercalation structure, intercalated films after being treated with different time in model 3 were prepared, and their electronic conductivities were visually displayed by using as wires to light the bubbles. As shown in Fig. 5a–c, with increasing the intercalation time, the $TiO_2$ NF film changed from white to blue and then to black, and more NPs were gradually generated on the NFs. At the same time, the bulb became brighter, indicating a higher conductivity (inset of Fig. 5a–c). Figure 5d, e showed the XRD patterns of the three samples with different intercalation intensities. The peaks of $D\text{-}Li_xTiO_{2-\delta}$ were obviously weakened and had a shift toward lower angles, implying that more $Li^+$-ions intercalated into the $TiO_2$ lattices with increasing time, which led to lattice distortion and rearrangement[41].

Then, the intercalation depth was characterized by using XPS depth profiling tests. As shown in Fig. 5f, an increase in Li content was observed according to the high-resolution spectra of Li 1s with the progress of $Li^+$-intercalation. Since lithium species might exist on the surface of $D\text{-}Li_xTiO_{2-\delta}$, the intercalated samples were washed with hydrochloric acid three times before the XPS tests. Furthermore, the high-resolution spectra of O 1s and Ti 2p were provided to evaluate the oxygen defects of $D\text{-}Li_xTiO_{2-\delta}$. As shown in Fig. 5g, the oxygen defect (cyan line) content of the sample that reacted for 30 min was significantly increased compared with the samples that reacted for 1 min and 1 s, indicating more oxygen defects generated in the $TiO_2$ lattice. The spectra of Ti 2p of the three samples showed two fitted peaks that ascribed to $Ti^{4+}$ and $Ti^{3+}$ species, and the $Ti^{3+}$ (cyan line) contents were gradually increased with the reaction time (Fig. 5h). Although the $Li^+$-intercalation created lots of vacancy defects, most of them existed on the $TiO_2$ surface[42]. As shown in Fig. 5i, it could not detect lithium elements in the $D\text{-}Li_xTiO_{2-\delta}$ NFs that were etched for 100 nm, indicating that the $Li^+$-intercalation mostly occurred on the NF surface.

In both model 1 and 3, the white $TiO_2$ NF films could turn black rapidly and both films exhibited high electronic conductivity of >40 S/m, but the black films showed different sensitivity to air. The black $Li_xTiO_{2-\delta}$ NF film prepared by model 3 gradually turned blue that could be kept for several months without further fading when exposed to the air. On the contrary, the black $Li_xTiO_{2-\delta}$ NF film prepared by model 1 could be stable only when it was stored in dry air (Supplementary Fig. 6a), and it would gradually change from black to gray white when exposed to the air (Supplementary Fig. 6b), indicating that the as-prepared black $Li_xTiO_{2-\delta}$ was metastable, and it achieved self-repair of surface defects. Simultaneously, the conductivity of the faded NF film decreased to 0. In model 1, when contact with the $TiO_2$ NF film, Li-metal quickly deprives O-atom on the $TiO_2$ crystal surface to form surface oxygen vacancies, thus leading to the rapid color change from white to black. However, these surface defects can introduce excessive charges to absorb water molecules in the air, then form hydroxyl species and fill the oxygen vacancy defects, resulting in the dynamic change of $Li_xTiO_{2-\delta}$ NF film from black to gray (Supplementary Fig. 6c)[43]. These results indicated that the intercalated $Li_xTiO_{2-\delta}$ was metastable, and it contained a low stable but high conductive black $TiO_{2-\delta}$ structure and a high stable but low conductive blue $Li_xTiO_2$ structure (Fig. 5j).

Interestingly, if heated the fresh metastable $Li_xTiO_{2-\delta}$ NF film in an inert atmosphere at 100 °C for 2 h, it exhibited excellent air stability and could be exposed to air for months without fading. The thermodynamics causes the transfer of surface oxygen vacancies into the $TiO_2$ lattice and forms stable defect structures[44]. From the UV–vis diffuse reflection spectroscopy (Fig. 5k), the black $Li_xTiO_{2-\delta}$ film shows the highest light absorption capacity, while both the gray $Li_xTiO_{2-\delta}$ and white $TiO_2$ films can hardly absorb light. By contrast, in model 3, $Li^+$-ions have priority to intercalate into the $TiO_2$ lattice with low energy under an electric field, rather than depriving the O-atom on the $TiO_2$ surface to form oxygen vacancies. This is due to that $Li^+$-ions are fixed under the action of intermolecular forces, thus forming stable blue $Li_xTiO_{2-\delta}$ NF structures. But with more $Li^+$-ions intercalation reactions, oxygen vacancies will also be formed and stable blue $Li_xTiO_{2-\delta}$NF will become metastable black $Li_xTiO_{2-\delta}$ NF. It is worth mentioning that the $Li^+$-intercalation could be precisely designed by controlling the intercalation time and paths, and both targeted metastable and stable $Li_xTiO_{2-\delta}$ materials could be synthesized with different models.

## Discussion

The energy barrier for $Li^+$-intercalation into $TiO_2$ is generally manipulated by thermodynamics and kinetics, but few studies have reported the intercalation pathways and the stability of the intercalated structures. The $Li^+$-ion intercalation in powdered metal oxides is difficult to observe and it is a challenge task to control the electronic structure and regulate the conductivity. Most reports focused on the structure changes of the intercalated metal oxide powders without considering the electronic conductivity changes. In this work, we studied the

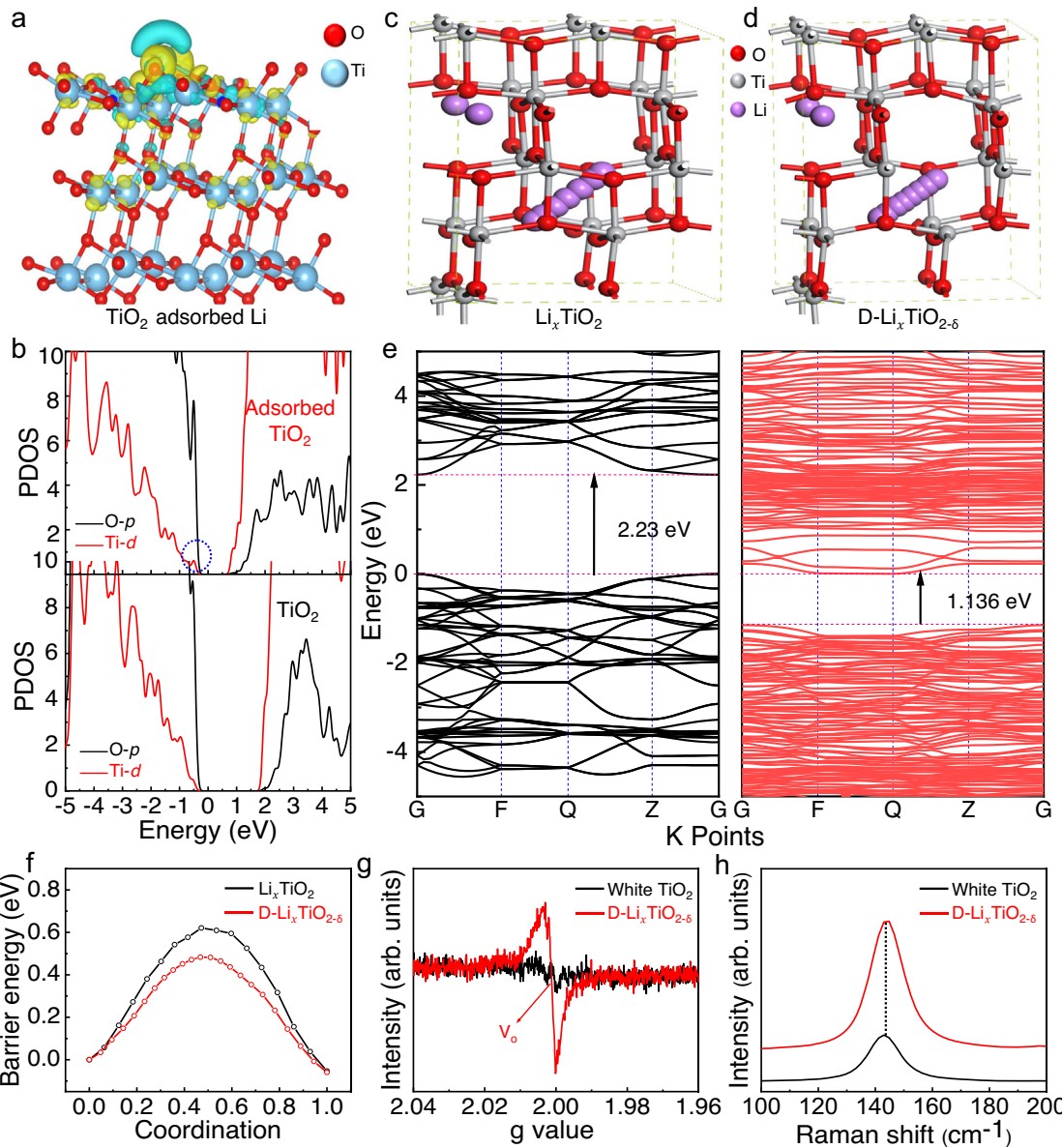

**Fig. 4 | Simulation and calculation characterizations. a** Optimized geometric structures and charge difference density mappings for the Li-adsorbed TiO₂. **b** PDOS plots of the initial TiO₂ and the intercalated TiO₂. **c, d** Li⁺-ions diffusion pathway for $Li_xTiO_2$ and $D\text{-}Li_xTiO_{2-\delta}$. **e** Band structure of $Li_xTiO_2$ and $D\text{-}Li_xTiO_{2-\delta}$. **f** Li⁺-ions diffusion energy barriers in $Li_xTiO_2$ and $D\text{-}Li_xTiO_{2-\delta}$. **g** EPR spectra and **h** Raman spectra of TiO₂ before and after the intercalation.

intercalation chemical reactions by using flexible TiO₂ NFs as a prototype. We found that the intercalation was positively related to the conductivity, and the stability of the intercalated structures were found to be closely related to the lattice oxygen vacancies, the concentration of Ti³⁺-species, and the different Ti³⁺-species that were created from different mechanisms[45,46]. This proof-of-concept was confirmed by designing a series of topological intercalation reactions using three different charge-driven models to control the concentrations and transfer pathways of the resident Li⁺-ions and electrons, which realized designable Li⁺-intercalation into TiO₂ NFs accompanying with a clear color change from white to blue and black of the intercalated $Li_xTiO_{2-\delta}$ NFs. The intercalation reaction would occur along the Li⁺-ion diffusion when the resident electrons were sufficient, and along the electron conduction if there were enough resident Li⁺-ions.

With the as-designed models, $Li_xTiO_{2-\delta}$ containing unstable but high conductive black TiO₂₋δ and stable but low conductive blue $Li_xTiO_2$ structures could be designed. It is worth noting that both

electrons and Li⁺-ions are necessary for intercalation. For example, the white TiO₂ NF films did not change if there was no wire connection or replacing the Li-metal sheet with a graphite rod in model 3 (Supplementary Fig. 7a), but the film quickly turned black if adding a 3.7 V DC power for the graphite electrode (Supplementary Fig. 7b), confirming that electron is necessary for the color change. In addition, if only immersed the bottom half of the TiO₂ NF film in the solvent, the soaked part changed from white to black, but the upper part in the air was still white (Supplementary Fig. 8), confirming that Li⁺-intercalation contributed to the color change. On the other hand, another two models 4 and 5 were built by adding a 3.7 V DC power for models 2 and 3 (Supplementary Fig. 9) to confirm the effect of electron concentration. For model 4, it still took ~24 h for the color change from white to black, confirming that the Li⁺-gating is the main limiting factor. By contrast, it only took 1 min for the color change from white to black in model 5 (Supplementary movie 4). The reduced time from 3 min to 1 min confirmed that the high electron concentration accelerated the intercalation reaction. The

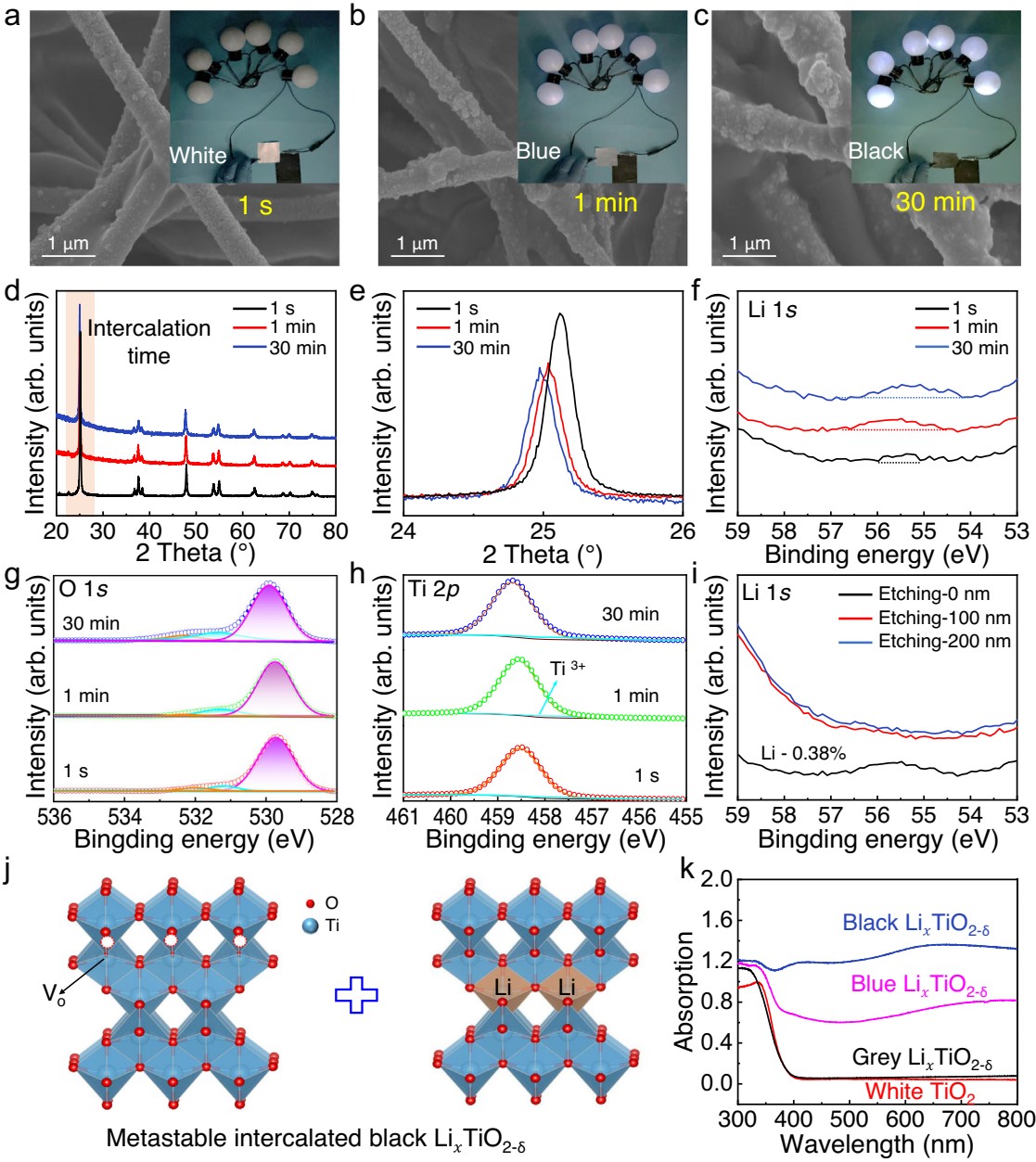

**Fig. 5 | Structures and structure stability of the conductive blue and black Li$_x$TiO$_{2-\delta}$.** SEM images of the white TiO$_2$, blue Li$_x$TiO$_{2-\delta}$ and black D-Li$_x$TiO$_{2-\delta}$ NFs treated by model 3 for **a** 1 s, **b** 1 min, and **c** 30 min, and visual displays of the conductivity of these NF films. **d**, **e** XRD patterns of D-Li$_x$TiO$_{2-\delta}$ after the different intercalation time. XPS spectra of **f** Li 1$s$, **g** O 1$s$ and **h** Ti 2$p_{3/2}$ of D-Li$_x$TiO$_{2-\delta}$ with different intercalation reaction times. **i** High-resolution XPS spectra of Li 1$s$ of the D-Li$_x$TiO$_{2-\delta}$ NFs after being deeply etched for 100 nm and 200 nm. **j** The generated metastable intercalated black Li$_x$TiO$_{2-\delta}$ that contained lots of surface oxygen vacancies and intercalated Li$^+$-ions. **k** UV–vis diffuse reflection spectroscopy of the four film samples.

TiO$_2$ NFs obtained from models 1, 4, and 5 exhibited similar rough structures as models 2 and 3, but the surface roughness of these intercalated Li$_x$TiO$_{2-\delta}$ was different (Supplementary Fig. 10). Moreover, from the XRD patterns (Supplementary Fig. 11), it is obvious that the peak shift of Li$_x$TiO$_{2-\delta}$ obtained from model 1 was the largest, suggesting that the divergent intercalation was rapid and intense. According to the TEM and FFT images (Supplementary Fig. 12), the lattice fringes in all models exhibited different degrees of distortions and the Raman spectra showed obvious blue shift and broadened $E_g$ peaks after the Li$^+$-intercalation (Supplementary Fig. 13) due to the lattice distortions[47]. The O 1$s$ spectra also confirmed that the intercalation caused the generation of oxygen vacancies, and the model with deep intercalation contained a higher oxygen defect (Supplementary Fig. 14).

Moreover, discussing in a broader sense, the proposed current-driven topological chemical synthesis strategy is universal and scalable. For one thing, by replacing the Li-metal with other conductive materials such as graphite in model 5, the intercalation reactions occurred normally, and the white TiO$_2$ NF film also quickly changed into black. However, under the same Li$^+$-ion concentration, the intercalation intensity corresponding to different opposite electrodes was different (Supplementary Fig. 15), indicating that the potential difference between TiO$_2$ and the counter electrodes would affect the intercalation reactions. In addition, replacing the DMAC solvent with other solvents with similar electron-pulling ability, such as toluene, also does not affect this experimental verification[48]. For another, in addition to Li$^+$-ions, Zn$^{2+}$-ions or other cation ions with higher valence could also be intercalated into TiO$_2$ NFs with the model 4 prototype (Supplementary Fig. 16). In

addition to the fundamental interests around the mechanisms of intercalation chemistry, the reported electrospinning method for the fabricated flexible ITMO NF films is also scalable, and we have prepared many flexible ITMO fabrics on large scale, such as $TiO_2$, $BaTiO_3$, $SnO_2$, $Nb_2O_5$ and so on[49–51]. These ITMO fabrics with superior softness like a napkin and controllable defects break the traditional perception of brittle and inert oxide ceramics.

Metal oxides with open interpenetrating tunnel network structures can be intercalated by cations with small atomic radius to form ITMO[52]. However, most studies focused on crystals, which increased the difficulty of studying the intercalation pathways. In this study, the flexible $TiO_2$ NF film had a porous structure formed by stacking randomly arranged NFs, which not only enhanced the continuity of electron conduction, but also improved the Li$^+$-ion transfer speed due to the siphoning effect of nanopore structures. This topological NF structure provided a platform for observing the directional transfer pathways of Li$^+$-ions and electrons, thus it is expected to control the intercalation reaction and prepare intercalation oxide compounds that meet the practical requirements. Studying the intercalation pathways and the control of the intercalation structures is of great significance to developing high-performance functional ITMO materials.

In summary, we have reported a visual topochemical synthesis strategy to study the intercalation chemical reaction pathways, processes, and structures from a perspective that is different from the traditional study on ITMO. The main findings of this study include the following three aspects. First, the intercalation reaction pathways are visualized in real-time, which has not been reported before. The topological NF structure provides a platform for observing the directional transfer pathways of Li$^+$-ions and electrons, thus it is expected to control the intercalation reaction and prepare intercalation oxides that meet the practical requirements. Second, the stability mechanism of the intercalation structures and the regulation mechanism of metal oxides' electronic conductivity by intercalation structures are clarified. An intercalation-based electron conduction pathway is established in the intercalation reactions, and the stability of the intercalated structures is found to be related to the lattice oxygen vacancies, the concentration, and the originality of Ti$^{3+}$-species. Third, the topochemical synthesis strategy is a universal method for the rapid synthesis of conductive metal oxide films on a large scale. The control of metal oxides' conductivity is of great significance to develop high-performance ITMO materials.

## Methods

### Fabrication of flexible topological $TiO_2$ NF films and the design of different charge-driven models

Flexible $TiO_2$ NF films were prepared with an electrospinning method followed by a high temperature calcination. Specifically, a clear solution was first obtained by dissolving PEO (Mw = 600,000) powder in a mixed solvent of acetic acid and ethanol with a weight ratio of 3:4. Then, titanium isopropoxide (TIP) was added into the clear solution. A transparent and homogeneous sol for electrospinning was obtained by ball-milling the solution for 30 min. Next, the spinning sol was injected at a speed of 10 ml/h and stretched by an applied voltage of 15 kV during the electrospinning process at room temperature and a humidity environment of 45% ± 2%. The precursor electrospun NF films were collected on a rotating collector, which was 150 mm from the spinneret. Then the precursor electrospun NF films were dried at a low temperature for 1 h to promote the volatilization of residual solvents. The last step was sintering the precursor films in a furnace at 600 °C for 2 h in air with a heating rate of 2 °C/min and then naturally cooling the furnace by turning off the heating button. Five charge-driven models were designed to verify the proposed strategy. Specifically, for model 2, 3, 4, and 5, $TiO_2$ NF films and circular Li-sheets were clamped and placed on both sides of the H-type electrolytic cells that contained different solvents. The other ends of the two alligator clips were

connected by a wire. The differences in these four models are listed as follows. The solvent was DMAc in models 2 and 4, and the solvent was a mixture of DMAc and LiClO$_4$ in models 3 and 5. In addition, there was an output power device with an applied voltage of 3.7 V in models 4 and 5, but a naked wire in models 2 and 3. On the other hand, in model 1, a naked Li-metal was directly connected with a $TiO_2$ NF film, and the latter was infiltrated with ~1 ml of DMAc. The experiments were recorded with a camera in real-time.

### Material characterization

The structures of $TiO_2$ and the intercalated $Li_xTiO_{2-\delta}$ NF films were characterized by field emission SEM (Hitachi S-4800) and HRTEM (JEM-2100F). The crystal and chemical structures were checked via Bruker XRD with Cu Kα radiation, XPS (PHI 5000C ESCA System) and Raman (LabRAM HR Evolution). UV–visible diffuse reflectance spectra were obtained by a spectrophotometer (Hitachi U-3900). The electronic conductivity was measured by an ST-2258C multifunction digital four-probe tester.

### Computational simulation

DFT calculations were performed by using the Vienna Ab-initio Simulation Package (VASP) to study the adsorption of Li$^+$-ions on {101} surface of $TiO_2$ and their electronic structures. The exchange–correlation interactions were described by generalized gradient approximation (GGA) with the functional Perdew–Burke–Ernzerhof (PBE)[53]. The cut-off energies for plane waves were set to be 500 eV, and the residual force and energy on each atom during structure relaxation were converged to 0.005 eV Å$^{-1}$ and $10^{-5}$ eV, respectively. The adsorption energy ($E_{ad}$) is defined as: $E_{ad} = E_{surf + Li} + E_{Li} - E_{surf}$, where $E_{surf + Li}$ is the total energy of $TiO_2$ (101) adsorbing lithium, $E_{Li}$ is the total energy of Li, $E_{surf}$ is the total energy of $TiO_2$ (101).

The calculations of Li migration were carried out under the scheme of spin-polarized DFT using CASTEP. Specifically, the Perdew–Burke–Ernzerhof exchange-correlation function within the generalized gradient approximation was employed to describe the exchange-correlation energy. Geometric convergence tolerances were set for a maximum force of 0.03 eV/A˚, a maximum energy change of $10^{-5}$ eV/atom, a maximum displacement of 0.001 A˚ and a maximum stress of 0.5 GPa. The sampling in the Brillouin zone was set with $3 \times 3 \times 1$ by the Monkhorst–Pack method. The diffusion of Li was investigated by searching the possible diffusion route and identifying the migration transition state with the lowest diffusion energy barrier. The diffusion energy barrier was the energy difference between the total energies of transition state and the initial structure. The transition state was searched by the generalized synchronous transit method implemented in the CASTEP code. The algorithm started from a linear synchronous transit (LST) optimization, and continued with a quadratic synchronous transit (QST) maximization process. Thereafter, the conjugate gradient (CG) minimization was conducted from the obtained LST/QST structure to refine the geometry of transition state. The LST/QST/CG calculations were repeated till a stable transition state was obtained.

## Data availability

The data supporting the findings from this study are available within the article, Supplementary Information, or Source data file. Source data are provided with this paper.

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

## Acknowledgements

This work is supported by the National Natural Science Foundation of
China for Distinguished Young Scholars (No. T2122009 to J.Y.), Shanghai
Rising-Star Program (20QA1400600 to J.Y.), and Fundamental Research
Funds for the Central Universities (2232021A-04 to J.Y.).

## Author contributions

J.Y. and X.Z. conceived and designed the project. Y.Z. conducted the
experimental and materials characterizations. J.Y. and Y.Z. wrote
this paper and all authors contributed to discussing and revising
the paper.

## Competing interests

The authors declare no competing interests.

## Additional information

**Supplementary information** The online version contains
supplementary material available at

Jianhua Yan.

**Peer review information** *Nature Communications* thanks Kun (Kelvin)
Fu, Hao Wang, and the other, anonymous, reviewer(s) for their con-
tribution to the peer review of this work. A peer review file is available.

