## [Peer Review File · Nature Communications]

REVIEWER COMMENTS

Reviewer #1 (Remarks to the Author):

The manuscript by Yan et al. reports a visual topochemical synthesis strategy to realize a rapid synthesis of conductive TiO₂ nanofiber films in a short time and at room temperature. The authors designed several different charge-driven models to intercalate Li⁺-ions into the TiO₂ lattice at different speeds, and the color changes of TiO₂ nanofiber films could be directly observed in different models, which are related to the intercalation pathways, structures, and conductivity of TiO₂. The method can also be extended to synthesize other metal oxide films. In general, this research provides a new perspective to study intercalation reactions, but some points should be clarified. Therefore, I recommend the publication on Nature Communications with minor revisions as follows.

1. Intercalation is an important reaction in the field of electrochemical energy storage and is an effective method to optimize the electronic structures of material. The authors obtained conductive TiO₂ nanofiber film through Li⁺-ion intercalation. What's the dynamics of the quick intercalation into TiO₂ nanofiber film at room temperature? What's the correlation between the conductivity of the TiO₂ nanofiber film and the Li⁺-ions intercalation? I recommend adding more explanations or experimental results.
2. The reaction principle of these models in this manuscript is like the charging and discharging reaction of Li-ion batteries. Both the intercalation and deintercalation behavior of Li⁺-ions will occur. The authors have studied the intercalation behavior of Li⁺-ions. Is the intercalation reaction reversible in these models? Have the authors studied the deintercalation of Li⁺-ions?
3. The authors stated that the topochemical synthesis method can be extended. In Figure S16, the Zn²⁺-ions with higher valence could also be intercalated into TiO₂ nanofiber film. Have the authors studied the intercalation of Li⁺-ions in other metal oxide nanofiber films?
4. The Raman spectra data are important to characterize the structure of the intercalated TiO₂, but the data of TiO₂ in Figure 4f seems incomplete. It is suggested to add the complete Raman spectra in the manuscript or supplementary materials.
5. The main findings of this study should be discussed and summarized in a more concise and comprehensive manner. For example, what's the advantages of using the flexible TiO₂ nanofiber film as the model? Does the nanofiber diameter influence the intercalation? In addition, oxide nanofibers are usually brittle and prone to fracture under bending state, I recommend adding more discussion on the flexibility of TiO₂ nanofiber film.

Reviewer #2 (Remarks to the Author):

General comment

Zhang et al. developed a visual topochemical synthesis strategy to control the intercalation pathways, structures, and conductivity, and realize the rapid synthesis of flexible conductive metal oxide films at room temperature.

The authors motivate this work by saying it is challenging to manipulate the intercalation structure and conductivity of metal oxides by cation intercalation. While true, it is not clear what, if any, are the advantages of their films. In fact, the lithiation of metal oxides is widely investigated and applied in practical production, which lacks novelty.

The authors claimed that this developed topochemical synthetic strategy can be extended to synthesize other conductive metal oxide films, which can be applied in fast-charge electrodes, electrochromic pattern designs, catalysis, etc. Unfortunately, none of the above were investigated in this paper. It is meaningless to make such making such statements without empirical evidence. This work focuses on conductivity modulation, while the strategy is an advance, it does not seem to be a significant breakthrough of interest to the broad readership of Nature Communications.

Specific comments

1. The first concern is how the author defines the x value in Li_xTiO_{2-δ}. How to control the thickness of the lithiation?

2. One of this manuscript's novelties is the lithiation methods. Please provide more DMAc Li complex reaction mechanisms on the cathode side. It would be clearer if the author could demonstrate why DMAc Li is chosen.
3. In the XRD pattern (Fig. 3j), no signals referring to $\text{Li}_x\text{TiO}_2\text{-}\delta$ can be found. More characterizations need to be provided.
4. The devices, such as batteries, should be demoed to prove the advantage of the samples.
5. The simulation part is less convincing and needs more solid investigations. What principle are the atomic structures of defective TiO_2 samples based on? The models shown in Figure 4b seem not accurate. Moreover, the kinetics simulations on the lithiation of TiO_2 may tell more essence.
6. Detailed parameters for preparing TiO_2 nanofiber films should be provided.

Reviewer #3 (Remarks to the Author):

This manuscript, entitled "Control of Metal Oxides' Electronic Conductivity Through Visual Intercalation Chemical Reactions" reported the rapid synthesis of flexible conductive TiO_2 by controlling the intercalation pathways of TiO_2 . Furthermore, the authors revealed that the intercalation reaction occurs along the diffusion of Li^+ ions when there is a sufficient presence of resident electrons. The intercalated TiO_2 nanofiber film, manipulated through intercalation pathways and intensity, is applicable in fast-charge electrodes, electrochromic pattern designs, catalysis, and other areas. These findings and approaches hold potential to provide valuable insights to researchers in this field. However, there are several issues that require addressing before publication. The detailed comments are as follows:

- The authors need to explain the reasons for the flexibility of the TiO_2 nanofiber film compared to conventional heat treating conditions.
- In Figure 5, the authors need to quantify the total lithium content and conductivity as a function of intercalation time.
- The authors need to provide data on the conductivity along the lithium intercalation path (model 1,2 and 3) when the amount of intercalated lithium remains the same.
- In Figure 5, the conductivity of the TiO_2 nanofiber film increases as the intercalation time. The authors need to explain this conductivity behavior when it exceeds 30 minutes.

RESPONSE TO REVIEWERS' COMMENTS

Note: All critiques are highly appreciated. The changes are **highlighted in blue** in the revised Manuscript and revised Supplementary Information for review.

Reviewer #1:

Overall comments: The manuscript by Yan et al. reports a visual topochemical synthesis strategy to realize a rapid synthesis of conductive TiO₂ nanofiber films in a short time and at room temperature. The authors designed several different charge-driven models to intercalate Li⁺-ions into the TiO₂ lattice at different speeds, and the color changes of TiO₂ nanofiber films could be directly observed in different models, which are related to the intercalation pathways, structures, and conductivity of TiO₂. The method can also be extended to synthesize other metal oxide films. In general, this research provides a new perspective to study intercalation reactions, but some points should be clarified. Therefore, I recommend the publication on Nature Communications with minor revisions as follows.

Response: *We thank the reviewer for the valuable comments. Based on the sincere suggestions, we have appropriately revised the manuscript and added analysis clarification and discussion to make the paper easier to follow. We hope that the reviewer finds our responses satisfactory. The responses to comments are listed below:*

Specific Comments:

Comment 1- Intercalation is an important reaction in the field of electrochemical energy storage and is an effective method to optimize the electronic structures of material. The authors obtained conductive TiO₂ nanofiber film through Li⁺-ion intercalation. What's the dynamics of the quick intercalation into TiO₂ nanofiber film at room temperature? What's the correlation between the conductivity of the TiO₂ nanofiber film and the Li⁺-ions intercalation? I recommend adding more explanations or experimental results.

Response: *Thanks for these valuable comments. Generally, the intercalated host materials have empty sites available for guests to be intercalated, which is very important to intercalation reactions. In this case, there is a limit to the size of the insertion atom that can be incorporated without breaking bonds in the oxide matrix host, and only elements with a small atomic radius, such as H and Li, can be inserted at ambient temperatures. As a host material, TiO₂ has open channels in the lattice and small Li⁺-ions can overcome lower barrier energy to intercalate into the lattice of TiO₂*

^[1]. For example, the anatase TiO₂ has a tetragonal crystal structure with empty octahedral sites that reversibly accommodate Li intercalation and extraction ^[2]. In model 1, the strong reductant Li-metal can directly reduce the TiO₂ in contact with it due to the difference in energy level barrier and the created active solvated Li/Li⁺ due to the interaction of solvent (DMAc) and Li can synchronously intercalate into the TiO₂ lattice. These active solvated Li/Li⁺ and solvated electrons(e_s⁻) are highly reactive nucleophiles that can accelerate the reduction of TiO₂ nanofiber film by Li-metal. In model 2, because of the potential difference, the lithium metal transfers electrons to the TiO₂ nanofiber film through the wire. At the same time, the solvated Li⁺@DMAc will transfer from the Li-metal side to the TiO₂ nanofiber film side due to the concentration difference to form a continuous Li⁺-ion diffusion channel, thus resulting in intercalation of Li⁺-ions. In model 3, Li⁺-ions are sufficient. If a conductive path is formed between the TiO₂ nanofiber film and Li-metal, Li⁺-ions can be spontaneously and rapidly intercalated into the TiO₂ nanofiber film at room temperature. The Li⁺-intercalation induces the reduction of adjacent Ti from +4 to +3 state corresponding to the reduction of the band gap ^[3]. After intercalation, the color of the white TiO₂ nanofiber film begins to turn blue or black, with abundant oxygen vacancies and Ti⁺³ defects, which are beneficial to the improvement of the conductivity of TiO₂ nanofiber film. Therefore, it can be quickly prepared flexible and conductive black TiO₂ nanofiber films through these models. We have added these discussions in the revised manuscript (**Lines 84-87; 95-97; 148-152**). We hope the reviewer is satisfied with our revision.

References

- [1] J. Shin, D. Samuelis, J. Maier, Sustained Lithium-Storage Performance of Hierarchical, Nanoporous Anatase TiO₂ at High Rates: Emphasis on Interfacial Storage Phenomena, *Adv. Funct. Mater.* **21**, 3464-3472 (2011).
- [2] A. A. Belak, Y. Wang, A. Van der Ven, Kinetics of Anatase Electrodes: The Role of Ordering, Anisotropy, and Shape Memory Effects. *Chem. Mater.* **24**, 2894-2898 (2012).
- [3] B. H. Meekins, P. V. Kamat, Got TiO₂ Nanotubes? Lithium-Ion Intercalation Can Boost Their Photoelectrochemical Performance. *ACS Nano* **3**, 3437-3446 (2009).

Comment 2- The reaction principle of these models in this manuscript is like the charging and discharging reaction of Li-ion batteries. Both the intercalation and deintercalation behavior of Li⁺-ions will occur. The authors have studied the intercalation behavior of Li⁺-ions. Is the intercalation reaction reversible in these models? Have the authors studied the deintercalation of Li⁺-ions?

Response: We thank the reviewer for this valuable comment. The intercalation of Li^+ -ions into TiO_2 nanofiber film is reversible. TiO_2 is a commonly used intercalation/deintercalation anode material for Li^+ -ion batteries. During the charge and discharge process, Li^+ -ions will be continuously intercalated and deintercalated in the TiO_2 lattice. In our models, the as-designed Li^+ -ion intercalation reaction is like the discharge reaction of a lithium-ion battery. Li^+ -ions are continuously intercalated in the TiO_2 lattice. As shown in **Figure R1**, if a reverse voltage is applied or energy is given, Li^+ -ions will be deintercalated from TiO_2 nanofiber film, and the blue or black TiO_2 nanofiber film could turn back to white. Therefore, under certain conditions, the intercalation reaction is reversible in these models. By controlling the intercalation time and the conditions of intercalation, the reaction can be easily controlled. We hope the reviewers are satisfied with our revision.

Figure R1: Reversible intercalation experiment of TiO_2 nanofiber film.

Comment 3- The authors stated that the topochemical synthesis method can be extended. In Figure S16, the Zn^{2+} -ions with higher valence could also be intercalated into TiO_2 nanofiber film. Have the authors studied the intercalation of Li^+ -ions in other metal oxide nanofiber films?

Response: We thank the reviewer for this valuable comment. This topochemical lithium intercalation is extensible. As shown in **Figure R2**, a white Nb_2O_5 nanofiber film and a graphite rod are immersed into the DMAc solvent containing 1M LiClO_4 , and then a voltage of 3.7 V was applied between the two electrodes. After a short period, it can be clearly seen that the Nb_2O_5 nanofiber film changes from white to black. The intercalation of Li^+ -ions induces the transition of Nb^{+5} to Nb^{4+} or even lower valence, which reduces the band gap width of Nb_2O_5 and produces a white-to-black color change^[1]. Similarly, lithium intercalation introduces oxygen vacancy defects in black Nb_2O_5 nanofiber film, which improves its conductivity. These results show the scalability of the topological synthesis method mentioned in the manuscript (**Lines 425-**

426). We hope that the reviewer finds our responses satisfactory and convincing.

Figure R2: Intercalation experiment of lithium in the Nb_2O_5 nanofiber film.

Reference

- [1] X. Lin, S. Xia, L. Zhang, Y. Zhang, S. Sun, Y. Chen, S. Chen, B. Ding, J. Yu, J. Yan, Fabrication of Flexible Mesoporous Black Nb_2O_5 Nanofiber Films for Visible-Light-Driven Photocatalytic CO_2 Reduction into CH_4 . *Adv. Mater.* **34**, 2200756 (2022).

Comment 4- The Raman spectra data are important to characterize the structure of the intercalated TiO_2 , but the data of TiO_2 in Figure 4f seems incomplete. It is suggested to add the complete Raman spectra in the manuscript or supplementary materials.

Response: We thank the reviewer for this valuable comment. In the supplementary materials, we have added the complete Raman spectra (**supplementary Figure S5**). The Raman spectra of $D\text{-Li}_x\text{TiO}_{2-\delta}$ NFs showed a similar configuration with the initial TiO_2 NFs, but a clear blue-shift of peak and a broadened E_g peak were observed, further confirming the robust intercalated $\text{Li}_x\text{TiO}_{2-\delta}$ structure and the structural evolution by nonstoichiometric measurements. We hope that the reviewer finds our responses satisfactory and convincing.

Comment 5- The main findings of this study should be discussed and summarized in a more concise and comprehensive manner. For example, what's the advantages of using the flexible TiO_2 nanofiber film as the model? Does the nanofiber diameter influence the intercalation? In addition, oxide nanofibers are usually brittle and prone to fracture under bending state, I recommend adding more discussion on the flexibility of TiO_2 nanofiber film.

Response: We highly appreciate the reviewer for this constructive comment. The main findings of this study include the following three aspects.

First, the intercalation reaction pathways are visualized in real-time, which has not been reported before. The topological nanofiber structure provides a platform for observing the directional transfer pathways of Li^+ -ions and electrons, thus it is expected to control the intercalation reaction and prepare intercalation oxides that meet the practical requirements. **Second**, the stability mechanism of the intercalation structures and the regulation mechanism of metal oxides' electronic conductivity by intercalation structures are clarified. An intercalation-based electron conduction pathway is established in the intercalation reactions, and the stability of the intercalated structures is found to be related to the lattice oxygen vacancies, the concentration and originality of Ti^{3+} -species. **Third**, the reported topochemical synthesis strategy is a universal method for the rapid synthesis of conductive metal oxide films on large scale. Many kinds of flexible conductive metal oxide films with tunable conductivity have been synthesized by controlling the intercalation time and pathways. The control of metal oxides' conductivity is of great significance to develop high-performance Intercalation chemistry-based transition metal oxides for appealing applications in fast-charge electrodes, electrochromic pattern designs, catalysis and so on.

In addition, it is reported that the polycrystalline electrospun TiO_2 nanofiber film is usually brittle due to its large grain size and pore defects ^[1]. We have solved this problem from two aspects. First, reducing the diameter of TiO_2 nanofibers and using the complex grain boundaries between grains to reduce the crack propagation and prevent fiber breakage. Second, reducing the internal pore defects of TiO_2 nanofibers. Usually, the fracture of TiO_2 nanofibers often occurs in the defect because the stress is easy to concentrate in the defect, resulting in crack propagation extension and fiber fracture ^[2]. In this manuscript, we used a ball milling method to prepare the spinning sol to form small grains and adopted a gradient calcination method to reduce the uncontrollable pore defects generated by polymer decomposition, thereby preparing the flexibility of TiO_2 nanofibers. We have added these discussions in the revised manuscript (**Line 195-199**). We hope that the reviewer finds our revisions satisfactory.

References

- [1] N. Thakur, N. Thakur, V. Bhullar, S. Sharma, A. Mahajan, K. Kumar, D. P. Sharma, D. Pathak, TiO_2 nanofibers fabricated by electrospinning technique and degradation of MO dye under UV light. *Crystalline Materials*, 236, 239-250 (2021).
- [2] Z. Zhang, Z. Yang, W. Qian, Y. Chen, Y. Xu, X. Xu, Q. Zhao, H. Li, Y. Zhao, H. Zhan, Achieving enhanced toughness of a nanocomposite coating by lattice distortion at the variable metallic oxide interface. *Materials & Design*, 224, 111316 (2022).

Reviewer #2:

Overall comments: Zhang et al. developed a visual topochemical synthesis strategy to control the intercalation pathways, structures, and conductivity, and realize the rapid synthesis of flexible conductive metal oxide films at room temperature. The authors motivate this work by saying it is challenging to manipulate the intercalation structure and conductivity of metal oxides by cation intercalation. While true, it is not clear what, if any, are the advantages of their films. In fact, the lithiation of metal oxides is widely investigated and applied in practical production, which lacks novelty. The authors claimed that this developed topochemical synthetic strategy can be extended to synthesize other conductive metal oxide films, which can be applied in fast-charge electrodes, electrochromic pattern designs, catalysis, etc. Unfortunately, none of the above were investigated in this paper. It is meaningless to make such making such statements without empirical evidence. This work focuses on conductivity modulation, while the strategy is an advance, it does not seem to be a significant breakthrough of interest to the broad readership of Nature Communications.

Response: *We thank the reviewer for these valuable comments, they are very helpful for us to improve this study. We have carefully revised this manuscript and added data to make it easier to follow. First, we agree with the reviewer that intercalation chemical reaction is a basic and important chemical reaction, which has always dominated the fields of electrochemical energy storage. Ion intercalation is an effective method to regulate the properties of crystal materials. Due to the small size of Li^+ -ions, it is easy to be intercalated into the host material. Therefore, the intercalation of Li^+ -ions in metal oxide materials has been extensively studied by researchers, especially in rechargeable lithium battery electrodes.*

*However, we want to claim that this study is different with the previous studies. The transport and accumulation of Li^+ -ions in metal oxide electrodes is a diffusion process, which is determined by the Li^+ -ions concentration and the internal structure of the electrode and is difficult to manipulate. **For one thing**, the traditional oxide electrodes are usually powdered, which require binders and conductive agents to prepare electrodes for battery assembly. The intercalation process of Li^+ -ions in the powdered electrode is difficult to observe and it is a challenge task to control the electronic structure and regulate the conductivity of metal oxides by cation intercalation. **For another**, most reports focused on the structure changes of the intercalated metal oxide powders, but without considering the electronic conductivity changes. We found that the intercalation is positively related to the conductivity. **In addition**, when the powder materials are modified by the electrochemical method, it is*

difficult to recycle the material due to the use of binders and conductive agents. **In this study**, we used a flexible TiO_2 nanofiber film as the host material for intercalation and visualized the process of intercalation reaction through the directional color change of the nanofiber film. The intercalation modification of the flexible TiO_2 films can be simply realized by using the battery-like method, but no need to assemble into a closed lithium battery. Using the unique electronic transmission mode of nanofibrous films, the structure of metal oxides can be regulated by controlling the electronic conduction path. These are the advantages of flexible nanofiber films and the innovation of this study. Moreover, we have verified that the proposed method can be used to prepare black or brown conductive BaTiO_3 , SnO_2 , Nb_2O_5 and fast-ionic conductor such as LLTO nanofiber films, Indicating the scalability of the proposed method (**Figure R2 and R3**). In addition, the advantages of the conductive black $\text{Li}_x\text{TiO}_{2-\delta}$ nanofiber film were also proved by testing the Li-ion batteries at a high current rate of 0.5 C (as shown in **Figure R12** that to reply to your another comment). Furthermore, according to your suggestions, we have assembled some devices including electrochromism and catalysis (**Figure R4-R5**), the black metal oxides indeed show great performances. Since we focus on the material synthesis in this paper, we have removed these overclaimed points in the revised manuscript (**Lines 26, 76, 439, and 456**)

Figure R3: The intercalated ceramic nanofiber films synthesized with the same method.

Figure R4: The as-prepared simple electrochromic device. After injecting a drop of DMAc/LiClO₄ liquid, the sealed TiO₂ nanofiber film quickly turns black.

Figure R5: The comparison of catalysis performance between the white Nb₂O₅ and the black Nb₂O_{5-x} for photocatalytic reduction of CO₂ into CO and CH₄. Obviously, the black catalyst films exhibited better selectivity of CH₄ conversion.

In a short summary, the main findings of this study include the following four aspects. **First**, the intercalation reaction pathways are visualized in real-time, which has not been reported before. The topological nanofiber structure provides a platform for observing the directional transfer pathways of Li⁺-ions and electrons, thus it is expected to control the intercalation reaction and prepare intercalation oxides that meet the practical requirements. **Second**, the stability mechanism of the intercalation structures and the regulation mechanism of metal oxides' electronic conductivity by intercalation structures are clarified. An intercalation-based electron conduction pathway is established in the intercalation reactions, and the stability of the intercalated structures is found to be related to the lattice oxygen vacancies, the concentration and originality of Ti³⁺-species. **Third**, the reported topochemical synthesis strategy is a universal method for the rapid synthesis of conductive metal oxide films on large scale. Many kinds of flexible conductive metal oxide films with tunable conductivity have been synthesized by controlling the intercalation time and pathways. The control of metal oxides' conductivity is of great significance to develop high-performance Intercalation chemistry-based transition metal oxides for appealing applications in fast-charge

electrodes, electrochromic pattern designs, catalysis and so on. **Fourth**, the method of preparing this flexible ceramic nanofiber is novel. Metal oxide ceramics are commonly hard and brittle materials, when they are bent they typically crack. The importance of this findings lies in that by carefully controlling the grain boundaries in the nanofibers, the TiO₂ ceramic displayed properties associated with both hard and soft matter. One of the popular methods for creating oxygen vacancies into oxide ceramics is using high-temperature gas reduction, but this method is not suitable for ceramics nanofibers since they are prone to creep fracture during the high temperature reduction. In this work, the black TiO₂ nanofibers also demonstrated excellent flexibility. The third one is that the material fabrication method is scalable, and with respect to the large-area fabrication, we have developed a pilot production machine in our lab (**Figure R6a**). With this machine, we can obtain continuous films with a large width of 120 cm (**Figure R6b**).

We have added these discussions in the revised manuscript (**Lines 84-87; 144-152; 195-203; 373-382**) and revised the abstract and introduction (**Lines 15-26; 31-35; 55-57; 75-76**) to make the paper easier to follow. We hope that the reviewer finds our responses satisfactory and convincing. The responses to comments are listed below:

Figure R6: (a) The scalable machine for fabricating large-scale ceramic nanofiber films. (b) A large piece of the soft ceramic nanofiber film.

Specific Comments:

Comment 1- The first concern is how the author defines the x value in Li_xTiO_{2-δ}. How to control the thickness of the lithiation?

Response: Thanks for this valuable comment. For the as-designed three current-driven models, there are two possible main reactions. One is to use the strong reducibility of Li-metal to reduce TiO₂, as shown in equation (1). The second is lithium intercalation into the TiO₂ lattice, as shown in Equation (2).

In the literature on lithium-TiO₂ batteries, x represents the depth of lithium insertion,

ranging from 0 to 1, which can be determined by the depth of discharge^[1]. In our model, the intercalation time is equal to the discharge depth of the traditional battery. As the intercalation time increases, the x value increases. Therefore, the size of x can be controlled by controlling the intercalation time. **We conducted constant voltage testing using Li-metal and TiO₂ nanofiber film as two counter electrodes.** Based on the current-time curve and the mass of TiO₂ nanofiber film, the charge transfer in Equation 2 reaction can be roughly calculated, thus estimating the relationship between the value of x in Li _{x} TiO₂ and the insertion time. As shown in **Figure R7**, Li⁺-ions continuously embed into TiO₂ over time and almost reached the maximum amount in 51 seconds. However, this is an ideal estimate. In the actual intercalation reactions, Li⁺-ions cannot be extensively embedded in the TiO₂ lattice, which can cause its structure to collapse. The theoretical lithium insertion capacity of anatase TiO₂ is 0.5 mol Li per 1 mol of anatase TiO₂, corresponding to the transformation of tetragonal TiO₂ into orthorhombic crystal system Li_{0.5}TiO₂^[2]. In the actual electrochemical intercalation, the maximal lithium content in anatase is in the range of 0.5-1 Li per Ti, depending on the experimental techniques^[3-4]. That is, the depth of lithium intercalation in TiO₂ will be stable. When the intercalation time continues to increase after reaching the maximal lithium content, the TiO₂ will be directly reduced by Li-metal, which is different from the battery discharge intercalation reaction. **Both the lithium intercalation and lithium reduction can lead to an increase in the conductivity of the TiO₂ nanofiber film.** In the manuscript, we assume that Li⁺-ions in the black TiO₂ has reached its maximum amount although this is not very accurate. We hope that the reviewer finds our revisions satisfactory and convincing.

Figure R7: The estimated nominal lithium concentration x in Li _{x} TiO₂.

References

- [1] B. J. Morgan, G. W. Watson, Role of Lithium Ordering in the Li _{x} TiO₂ Anatase → Titanate

Phase Transition, J. Phys. Chem. Lett. **2**, 1657-1661 (2011).

- [2] G. Nuspl, K. Yoshizawa, T. Yamabe, *Lithium intercalation in TiO₂ modifications*, *J. Mater. Chem. A*, **7**, 2529-2536 (1997).
- [3] T. Ohsuku, T. Hirai, *An electrochromic display based on titanium dioxide*, *Electrochim. Acta*, **27**, 1263-1266 (1982).
- [4] D. Bi, J. Wang, Y. Sun, Z. Liao, *A study of anatase-TiO₂ as cathode material for ambient temperature lithium secondary cells.* *Proc. Electrochem. Soc.* **80**, 245-260(1980).

Comment 2- One of this manuscript's novelties is the lithiation methods. Please provide more DMAc Li complex reaction mechanisms on the cathode side. It would be clearer if the author could demonstrate why DMAc Li is chosen.

Response: *Thanks for this valuable comment. We have consulted some classic books and papers to make clear of the reaction mechanisms between DMAc and Li. We think that DMAc plays two main roles in these models. Firstly, the TiO₂ nanofiber film shows good wettability with DMAc, which can effectively increase the contact areas between lithium metal and the TiO₂ nanofiber film. As shown in **Figure R8**, the TiO₂ nanofiber film is rapidly infiltrated by DMAc within 0.94 seconds. Due to the spontaneous chemical reactions, the TiO₂ nanofiber film connected with lithium can be reduced. It can be observed that the white TiO₂ nanofiber film changed from white to black quickly. This is consistent with the experimental phenomenon of model 1. Secondly, we think that some solvents, such as DMAc that contains C=O functional groups, can pull or lock electrons and active solvated Li/Li⁺ and solvated electrons (e_s⁻) can be formed when soaking Li-metal into these solvents. These solvated species are highly reactive nucleophiles that can accelerate the reduction of TiO₂ nanofiber film by Li-metal. There are literature report that alkali metals can dissolve in liquid ammonia and generate ammonia metal cations (M(NH₃)_mⁿ⁺) and ammonia electron (e(NH₃)_x⁻) [1-2]. This solution in a metastable state has strong reducibility. Similar to this phenomenon, electrons from Li-metal will be trapped by DMAc with electron pulling ability, producing highly reactive "e_s⁻". the DMAc solvent essentially acts as a phase transfer reagent, continuously transferring electrons from solid Li-metal to the liquid phase [3]. Due to the good wettability of TiO₂ nanofiber film to DMAc solvent, the TiO₂ nanofiber film can contact a large number of e_s⁻ and be reduced (**Figure R9**). At the same time, Li⁺-ions were intercalated into TiO₂, and the color of the TiO₂ nanofiber film changed, which is consistent with the experimental results of model 2. Based on the above reaction mechanism, we chose DMAc solvent for the lithiation of TiO₂ nanofiber film.*

In addition, we find that other solvents with electron-withdrawing groups (*N,N*-Dimethylformamide (DMF), 1-Methyl-2-pyrrolidinone (NMP), 1,2-dimethoxy-ethane (DME) et al.) also have the same effect, but due to differences in electron-withdrawing ability, the rate of lithium reduction reactions varies greatly. We also have added the complex reaction mechanisms in the revised manuscript (Line 84-87, Line 148-152). We hope that the reviewer finds our revisions satisfactory and convincing.

Figure R8: Dynamic behavior of DMAc on TiO₂ nanofiber film.

Figure R9: Schematic diagram of the interaction between DMAc and lithium.

References

- [1] T. Buttersack, P. E. Mason, R. S. McMullen, H. C. Schewe, T. Martinek, K. Brezina, M. Crhan, A. Gomez, D. Hein, G. Wartner, R. Seidel, H. Thürmer, O. Marsalek, B. Wintwe, S. E. Bradforth, P. Jungwirth, Photoelectron spectra of alkali metal-ammonia microjets: From blue electrolyte to bronze metal. *Science*, **368**, 1086-1091 (2020).
- [2] J. Burrows, S. Kamo, K. Koide, Scalable Birch reduction with lithium and ethylenediamine in tetrahydrofuran, *Science*, **374**, 741-746 (2021).
- [3] X. Chen, L. Qin, J. Sun, S. Zhang, D. Xiao, Y. Wu, Phase Transfer-Mediated Degradation of Ether-Based Localized High-Concentration Electrolytes in Alkali Metal Batteries, *Angew. Chem. Int. Ed.* **134**, e202207018 (2022).

Comment 3- In the XRD pattern (Fig. 3j), no signals referring to $\text{Li}_x\text{TiO}_{2-\delta}$ can be found. More characterizations need to be provided.

Response: We thank the reviewer for this valuable comment. We have added some more characterizations to confirm the formation of $\text{Li}_x\text{TiO}_{2-\delta}$. The reaction formula of Li^+ intercalation into TiO_2 nanofiber is as follows:

where x represents the depth of lithium intercalation, ranging from 0 to 1. When x is relatively small, the XRD diffraction peak of Li_xTiO_2 is similar to the initial TiO_2 . In our models, the intercalation time is similar to the discharge depth of the traditional Li^+ -ions batteries. As the intercalation time increases, the value of x increases, accompanying by a change in the TiO_2 nanofiber film from white to blue and then black. Figure 3j showed the XRD patterns of the white TiO_2 and black intercalated $\text{Li}_x\text{TiO}_{2-\delta}$ nanofiber films in different models. As described in the manuscript, the white TiO_2 nanofiber films in different models can become black after 3 min of lithium intercalation. Although the color of TiO_2 nanofiber films becomes black, no obvious signal of $\text{Li}_x\text{TiO}_{2-\delta}$ is observed due to the short intercalation time and less lithium intercalation. The XRD diffraction peak of $\text{Li}_x\text{TiO}_{2-\delta}$ is similar to the initial TiO_2 . But the intensity of the XRD diffraction peak of the intercalated $\text{Li}_x\text{TiO}_{2-\delta}$ becomes weaker, and the dispersion degree of the peak increases, which indicates that impurities appear in the TiO_2 lattice, and also indicates the intercalation of lithium (**Figure R10**). It is worth noting that when we further extend the lithium intercalation time, the XRD peak of the intercalated $\text{Li}_x\text{TiO}_{2-\delta}$ begins to show a slight phase transition (**Figure R11**). These results indicate that lithium is successfully intercalated into TiO_2 nanofiber films. We hope that the reviewer finds our revisions satisfactory and convincing.

Figure R10: XRD patterns of the white TiO_2 , black $\text{Li}_x\text{TiO}_{2-\delta}$ and $\text{D-Li}_x\text{TiO}_{2-\delta}$ nanofiber films.

Figure R11: XRD patterns of the white TiO₂, and black Li_xTiO_{2-δ} nanofiber films with a long-time intercalation of lithium.

Comment 4- The devices, such as batteries, should be demoed to prove the advantage of the samples.

Response: We thank the reviewer for this valuable comment. Lithium intercalation does affect the electrochemical performance of the TiO₂ nanofiber film. The electrochemical properties of pure TiO₂ nanofiber film and lithiated TiO₂ nanofiber film (Li_xTiO_{2-δ}) were investigated by assembling the CR2032 coin-type cells with Li-metal as the counter electrode. These flexible nanofiber films can be used as independent electrodes, which can effectively improve the gravimetric capacity and gravimetric energy storage of batteries ^[1]. As shown in **Figure R12**, the Li_xTiO_{2-δ} nanofiber film delivers a higher reversible capacity and better long-term stability than the pure TiO₂ nanofiber film at a high current rate of 0.5 C. According to the literature, TiO₂ possesses intercalation pseudocapacitive charge storage behavior that occurs when ions intercalate into the tunnels or layers of the active materials accompanied by a faradaic charge transfer with no crystallographic phase transition, which helps to improve the rapid charge storage ^[2]. Moreover, the lithiation induces the reduction of adjacent Ti from the +4 to the +3 state, corresponding to the reduction of the band gap. Thus, the Li_xTiO_{2-δ} nanofiber film demonstrates good electronic conductivity. The improvement of these electronic structures is conducive to enhancing the electrochemical performance of the materials, which enables the Li_xTiO_{2-δ} nanofiber film as a promising anode. We hope that the reviewer finds our revisions satisfactory and convincing.

Figure R12: Long-term cycling performance of the white TiO_2 and black $\text{Li}_x\text{TiO}_{2-\delta}$ nanofiber anode films.

References

- [1] W. Yuan, B. Wang, H. Wu, M. Xiang, Q. Wang, H. Liu, Y. Zhang, H. Liu, S. Dou, A flexible 3D nitrogen-doped carbon foam@ CNTs hybrid hosting TiO_2 nanoparticles as free-standing electrode for ultra-long cycling lithium-ion batteries. *J. Power Sources*, **379**, 10-19 (2018).
- [2] C. Chen, Y. Wen, X. Hu, X. Ji, M. Yan, L. Mai, P. Hu, B. Shan, Y. Huang, Na^+ intercalation pseudocapacitance in graphene-coupled titanium oxide enabling ultra-fast sodium storage and long-term cycling, *Nat. Commun.* **6**, 6929 (2015).

Comment 5- The simulation part is less convincing and needs more solid investigations. What principle are the atomic structures of defective TiO_2 samples based on? The models shown in Figure 4b seem not accurate. Moreover, the kinetics simulations on the lithiation of TiO_2 may tell more essence.

Response: Thanks for these valuable comments. According to your suggestions, we have added some more simulation calculations to make it easier to follow. In the revised manuscript, we have modified Figure 4 and added more explanations on the structure and calculations. **Figure R13a** is the optimized geometric structures and charge difference density mappings for the adsorption of Li on the TiO_2 {101} surface, which is used to explain the charge transfer between TiO_2 nanofiber film and Li-metal. The blue surfaces represent electron accumulation, which indicates the substantial electron transfer from the Li atom to the Ti atom, confirming that electrons can tunnel the Ti-O bond and then move forward to the next Ti-O bond to form a continuous conduction path along the -Ti-O-Ti-O- channel. The projected density of states (PDOS) of TiO_2 before and after adsorption of lithium is shown in **Figure R13b**. The calculated PDOS

plots exhibited that a new electron-filled band gap state consisting of Ti 3d sat on the left side of the Fermi Level, which explained the existence of the Ti^{3+} species that owe to the partial reduction of Ti sites on the TiO_2 nanofiber surface^[1].

To further illustrate the structure of lithium intercalated TiO_2 and the kinetics of Li^+ -ions migration, we add some kinetics simulations on the lithiation of TiO_2 in the revised manuscript. In our models, Li^+ -ions are first intercalated in TiO_2 to form Li_xTiO_2 . Then Li^+ -ions continue to insert, and $Li_xTiO_{2-\delta}$ can be obtained at the same time. There are vacant octahedral and tetrahedral sites in TiO_2 , that can accommodate Li^+ -ions. It is reported that lithium insertion proceeds into the octahedral sites of TiO_2 ^[2]. Based on this, we established the structure of Li_xTiO_2 and $Li_xTiO_{2-\delta}$ containing oxygen vacancies and calculated the lithium diffusion pathway from one stable site to the adjacent stable one (**Figure R13c-d**). After the incorporation of Li^+ -ions, the $Li_xTiO_{2-\delta}$ containing oxygen vacancies demonstrated a smaller band gap (2.23 eV), as compared with the Li_xTiO_2 (1.136 eV), both are smaller than the pristine TiO_2 (**Figure R13e**). Usually, the small band gap is beneficial to the generation of intrinsic electrons or holes^[3]. The narrow band gap of $Li_xTiO_{2-\delta}$ suggests the reduction of Ti from +4 to +3 state and the enhanced electron transfer ability in TiO_2 induced by lithiation^[4]. The energy profiles along the transport pathway are depicted in **Figure R13f**. For Li_xTiO_2 , the energy barrier for lithium diffusion is calculated to be 0.62 eV, which is higher than that of $Li_xTiO_{2-\delta}$ (0.48 eV), indicating the rapid intercalation of Li^+ -ions. We have added the kinetics simulations on the lithiation of TiO_2 in Figure 4. We have added these results in the main manuscript text (**Lines 276-293**) and we hope that the reviewer finds our responses satisfactory and convincing.

References

- [1] Y. Cao, P. Zhou, Y. Tu, Z. Liu, B. Dong, W. Azad, Ma, A. Modification of TiO_2 Nanoparticles with organodiboron molecules inducing stable surface Ti^{3+} complex. *iScience* **20**, 195-204 (2019).
- [2] M. V. Koudriachova, S. W. d. Leeuw, Orthorhombic distortion on Li intercalation in anatase, *Phys. Rev. B*, **69**, 054106 (2024).
- [3] F. Zhou, K. Kang, T. Maxisch, G. Ceder, D. Morgan, The electronic structure and band gap of $LiFePO_4$ and $LiMnPO_4$. *Solid State Commun.* **132**, 181-186 (2004).
- [4] L. Zhang, X. Zhang, G. Tian, Q. Zhang, M. Knapp, H. Ehrenberg, G. Chen, Z. Shen, G. Yang, L. Gu, F. Du, Lithium lanthanum titanate perovskite as an anode for lithium ion batteries, *Nat Commun.* **11**, 3490 (2020).

Figure R13. Simulation and calculation characterizations. (a) Optimized geometric structures and charge difference density mappings for the Li-adsorbed TiO_2 . (b) PDOS plots of the initial and the intercalated TiO_2 . (c-d) Li^+ -ions diffusion pathway for Li_xTiO_2 and $\text{D-Li}_x\text{TiO}_{2-\delta}$. (e) Band structure of Li_xTiO_2 and $\text{D-Li}_x\text{TiO}_{2-\delta}$. (f) Li^+ -ions diffusion energy barriers in Li_xTiO_2 and $\text{D-Li}_x\text{TiO}_{2-\delta}$. (g) EPR spectra and (h) Raman spectra of TiO_2 before and after the intercalation.

Comment 6- Detailed parameters for preparing TiO_2 nanofiber films should be provided.

Response: We thank the reviewer for this valuable comment. We have added the detailed parameters for preparing TiO_2 nanofiber films in the method section of the revised manuscript (Lines 463-469). We hope that the reviewer finds our revisions satisfactory and convincing.

Reviewer #3:

Overall comments: This manuscript, entitled "Control of Metal Oxides' Electronic Conductivity Through Visual Intercalation Chemical Reactions" reported the rapid synthesis of flexible conductive TiO₂ by controlling the intercalation pathways of TiO₂. Furthermore, the authors revealed that the intercalation reaction occurs along the diffusion of Li⁺ ions when there is a sufficient presence of resident electrons. The intercalated TiO₂ nanofiber film, manipulated through intercalation pathways and intensity, is applicable in fast-charge electrodes, electrochromic pattern designs, catalysis, and other areas. These findings and approaches hold potential to provide valuable insights to researchers in this field. However, there are several issues that require addressing before publication. The detailed comments are as follows:

Response: *We thank the reviewer for the valuable comments on our manuscript and the comments are very helpful in improving our manuscript. Based on the sincere suggestions, we have carefully revised the manuscript and added analysis clarification to make the paper easier to follow. We hope that the reviewer finds our responses satisfactory and convincing. The responses to comments are listed below:*

Specific Comments:

Comment 1- The authors need to explain the reasons for the flexibility of the TiO₂ nanofiber film compared to conventional heat-treating conditions.

Response: *Thanks for this valuable comment. Metal oxide ceramics are commonly hard and brittle materials, when they are bent, they typically crack. Usually, TiO₂ suffers from intrinsic brittleness and hardness and is easy to fracture. To solve this problem, developing one-dimensional TiO₂ nanofibers by electrospinning is expected to withstand larger external stress through the axial accumulation of local micro-deformations owing to the advantages such as large aspect ratio and good continuity, thus significantly improving the toughness. However, polycrystalline electrospun TiO₂ nanofiber film prepared by conventional heat treating is usually brittle due to its large grain size and pore defects^[1-2]. It is reported that reducing grain size and pore defects is beneficial for modifying the mechanical properties of oxide ceramic nanofibers^[3]. In this manuscript, we improved electrospinning conditions to obtain flexible TiO₂ nanofibers. For one thing, we ball milled the spinning sol to promote the formation of small grains and obtained a dense nanofiber structure at a low sintering temperature. For another, we adopted a gradient calcination method to reduce the uncontrollable pore defects generated by polymer decomposition, thereby enhancing the flexibility of*

TiO₂ nanofibers. As mentioned in the method section (**Line 471-477**), we calcined the precursor nanofibers at a low temperature for a while to promote the polymer molecular chain vibration and solvent evaporation, thereby reducing the pore defects. Through these improved methods, flexible TiO₂ nanofibers films were obtained.

In addition, one of the popular methods for creating oxygen vacancies into oxide ceramics is using high-temperature gas reduction, but this method is not suitable for ceramics nanofibers since they are prone to creep fracture during the high temperature reduction. In this work, the black TiO₂ nanofibers also demonstrated excellent flexibility. Moreover, the material fabrication method is scalable, and with respect to the large-area fabrication, we have developed a pilot production machine in our lab (**Figure R6a**, DXES-V machine, SOF Nanotechnology Co., Ltd.). With this machine, we can obtain continuous films with a large width of 120 cm (**Figure R6b**). We have added these results in the main manuscript text (**Lines 195-199, Supplementary Figure S4**) and we hope that the reviewer finds our responses satisfactory and convincing.

References

- [1] N. Thakur, N. Thakur, V. Bhullar, S. Sharma, A. Mahajan, K. Kumar, D. P. Sharma, D. Pathak, TiO₂ nanofibers fabricated by electrospinning technique and degradation of MO dye under UV light. *Crystalline Materials*, 236, 239-250 (2021).
- [2] O. Secundino-Sánchez, J. Diaz-Reyes, J. Aguila-López, J. F. Sánchez-Ramírez, Crystalline phase transformation of electrospinning TiO₂ nanofibres carried out by high temperature annealing. *J. Mol. Struct.* **1194**, 163-170 (2019).
- [3] J. Yan, Y. Han, S. Xia, X. Wang, Y. Zhang, J. Yu, B. Ding, Polymer Template Synthesis of Flexible BaTiO₃ Crystal Nanofibers, *Adv. Funct. Mater.* **29**, 1907919 (2019).

Comment 2- In Figure 5, the authors need to quantify the total lithium content and conductivity as a function of intercalation time.

Response: We highly appreciate the reviewer for the constructive comment. Figure 5 is a further explanation of the lithium intercalation reaction with model 3 as an example. There are two possible reaction paths during the intercalation reaction from 0 to 30 min. The first is the intercalation of Li⁺-ions.

The second is the reduction of TiO₂ nanofiber film by Li-metal.

Both reactions will cause the generation of Ti³⁺ and oxygen vacancy defects in TiO₂,

which is beneficial to improve conductivity and finally obtain conductive black Li_xTiO_2 nanofiber films.

In the literature on lithium- TiO_2 batteries, x represents the depth of lithium insertion, ranging from 0 to 1, which can be determined by the depth of discharge ^[1]. In our model, the intercalation time is equal to the discharge depth of the traditional battery. As the intercalation time increases, the x value increases. Therefore, the size of x can be controlled by controlling the intercalation time. **We conducted constant voltage testing using Li-metal and TiO_2 nanofiber film as two counter electrodes.** Based on the current-time curve and the mass of TiO_2 nanofiber film, the charge transfer in Equation 2 reaction can be roughly calculated, thus estimating the relationship between the value of x in Li_xTiO_2 and the insertion time. As shown in **Figure R7**, Li^+ -ions continuously embed into TiO_2 over time and almost reached the maximum amount in 51 seconds. However, this is an ideal estimate. In the actual intercalation reactions, Li^+ -ions cannot be extensively embedded in the TiO_2 lattice, which can cause its structure to collapse. The theoretical lithium insertion capacity of anatase TiO_2 is 0.5 mol Li per 1 mol of anatase TiO_2 , corresponding to the transformation of tetragonal TiO_2 into orthorhombic crystal system $\text{Li}_{0.5}\text{TiO}_2$ ^[2]. In the actual electrochemical intercalation, the maximal lithium content in anatase is in the range of 0.5-1 Li per Ti, depending on the experimental techniques ^[3-4]. That is, the depth of lithium intercalation in TiO_2 will be stable. When the intercalation time continues to increase after reaching the maximal lithium content, the TiO_2 will be directly reduced by Li-metal, which is different from the battery discharge intercalation reaction. **Both the lithium intercalation and lithium reduction can lead to an increase in the conductivity of the TiO_2 nanofiber film.** In the manuscript, we assume that Li^+ -ions in the black TiO_2 has reached its maximum amount although this is not accurate.

Figure R7: The estimated nominal lithium concentration x in Li_xTiO_2 .

In addition, we conducted conductivity tests on samples with Li^+ -ions intercalation at different times. As shown in **Figure R14**, when $0 < t < 3 \text{ min}$, the TiO_2 nanofiber film undergoes a color change from white to blue and then to black, and the conductivity increases from almost 0 to about 40 S/m. As the intercalation time increases ($3 \text{ min} < t < 30 \text{ min}$), the lithium content inside the $\text{Li}_x\text{TiO}_{2-\delta}$ nanofiber film continues to increase, indicating the increase of x . However, the conductivity of the black $\text{Li}_x\text{TiO}_{2-\delta}$ nanofiber film increased slowly, only increasing from 40 S/m to 50 S/m. Moreover, when $t > 30 \text{ min}$, the conductivity of the black $\text{Li}_x\text{TiO}_{2-\delta}$ nanofiber film almost unchanged and remained at $\sim 50 \text{ S/m}$. These results indicate that the depth of lithium intercalation and lithium reduction is limited. When intercalation or reduction reaches a certain level, the conductivity of the intercalated TiO_2 nanofiber film will not increase again, which is consistent with the literature results^[2]. We hope that the reviewer finds our responses satisfactory and convincing.

Figure R14: Changes in conductivity of $\text{Li}_x\text{TiO}_{2-\delta}$ nanofiber films over time.

References

- [1] H. Li, S. Wang, M. Wang, Y. Gao, J. Tang, S. Zhao, H. Chi, P. Zhang, J. Qu, F. Fan, C. Li, Enhancement of Plasmon-Induced Photoelectrocatalytic Water Oxidation over Au/TiO_2 with Lithium Intercalation, *Ange. Chem. Int. Ed.* **134**, e202204272 (2022).
- [2] J. Yan, Y. Zhang, Y. Zhao, J. Song, S. Xia, S. Liu, J. Yu, B. Ding, Transformation of oxide ceramic textiles from insulation to conduction at room temperature." *Sci. Adv.* **6**, eaay8538 (2020).

Comment 3- The authors need to provide data on the conductivity along the lithium intercalation path (model 1, 2 and 3) when the amount of intercalated lithium remains the same.

Response: We thank the reviewer for the valuable comment. According to **Figure 2**,

due to the difference in charge gating, the Li^+ -ions intercalation paths of the white TiO_2 nanofiber films are different in model 1, 2, and 3, thus resulting in different intercalation rates and inversion. However, the Li^+ -intercalation in all models led to the reduction of Ti^{4+} to Ti^{3+} , lattice expansion, and the creation of oxygen vacancy in TiO_2 crystals, resulting in real-time color changes of the film from white to blue and then black. As shown in **Figure R15**, after a long enough time, TiO_2 nanofiber films in all models turned black. In the manuscript, we assume that Li^+ -ions in black TiO_2 has reached its maximum amount. The intercalation of Li^+ -ions in TiO_2 nanofibers is limited along with the interaction time (as shown in Figure R14). Therefore, we tested the conductivity of black $\text{Li}_x\text{TiO}_{2-\delta}$ nanofiber films obtained by long-term intercalation. The conductivity of these three black $\text{Li}_x\text{TiO}_{2-\delta}$ nanofiber films prepared by the three different models is about 50 S/m, indicating that different models can change the intercalation rate, but cannot change the kinetic nature of Li^+ -ions intercalation. We hope that the reviewer finds our responses satisfactory and convincing.

Figure R15: Optical photos of $\text{Li}_x\text{TiO}_{2-\delta}$ nanofiber films after lithium intercalation for long enough in the three models.

Comment 4- In Figure 5, the conductivity of the TiO_2 nanofiber film increases as the intercalation time. The authors need to explain this conductivity behavior when it exceeds 30 minutes.

Response: We thank the reviewer for this valuable comment. According to the results of **Figure 1** and **Figure 5**, we can see that with the intercalation of Li^+ -ions in TiO_2 , the color of white TiO_2 nanofiber film changes from white to blue and then to black, and the deepening of the color indicates the increase of Li^+ -ions insertion. With the intercalation of Li^+ -ions, the conductivity of the TiO_2 nanofiber film increases gradually. However, the intercalation of Li^+ -ions in TiO_2 nanofibers is limited (as shown in **Figure R14**). In the beginning, Li^+ -ions can be quickly inserted into the TiO_2 lattice by overcoming the lower energy barrier. However, with the insertion of Li^+ -ions,

the migration of Li⁺-ions in the TiO₂ requires much energy, and the intercalation of Li⁺-ions also needs to overcome a larger energy barrier. Therefore, we think that after a certain period time, the conductivity of the black intercalated TiO₂ nanofibers film will not continue to increase with the extension of the intercalation time. That is, when the TiO₂ nanofibers film completely blackened, with the extension of time (more than 30 min), its conductivity is almost no longer improved. We hope that the reviewer finds our responses satisfactory and convincing.

REVIEWERS' COMMENTS

Reviewer #1 (Remarks to the Author):

Authors have addressed my questions/concerns. The current version could be considered for publication.

Reviewer #2 (Remarks to the Author):

The response is comprehensive and well-organized, significantly improving and enriching the manuscript. I have no further concerns and suggest its publication.

Reviewer #3 (Remarks to the Author):

The authors have satisfactorily addressed the reviewer's comments. The revised manuscript is acceptable for publication in this journal.

RESPONSE TO REVIEWERS' COMMENTS

Note: All critiques are highly appreciated. The changes are **highlighted in blue** in the revised Manuscript for review.

Reviewer #1:

Overall comments: Authors have addressed my questions/concerns. The current version could be considered for publication.

Response: *We thank the reviewer for the comment.*

Reviewer #2:

Overall comments: The response is comprehensive and well-organized, significantly improving and enriching the manuscript. I have no further concerns and suggest its publication.

Response: *We thank the reviewer for the comment.*

Reviewer #3:

Overall comments: The authors have satisfactorily addressed the reviewer's comments. The revised manuscript is acceptable for publication in this journal.

Response: *We thank the reviewer for the comment on our manuscript.*